# Assessment of the Life Cycle of a Wind and Photovoltaic Power Plant in the Context of Sustainable Development of Energy Systems

**DOI:** 10.3390/ma15217778

**Published:** 2022-11-04

**Authors:** Katarzyna Piotrowska, Izabela Piasecka, Zbigniew Kłos, Andrzej Marczuk, Robert Kasner

**Affiliations:** 1Faculty of Mechanical Engineering, Lublin University of Technology, 20-618 Lublin, Poland; 2Faculty of Mechanical Engineering, Bydgoszcz University of Science and Technology, 85-796 Bydgoszcz, Poland; 3Faculty of Civil Engineering and Transport, Poznań University of Technology, 60-965 Poznań, Poland; 4Faculty of Production Engineering, University of Life Sciences in Lublin, 20-950 Lublin, Poland

**Keywords:** wind power plant, photovoltaic power plant, life-cycle assessment (LCA), ReCiPe 2016, sustainable development

## Abstract

The conversion of kinetic energy from wind and solar radiation into electricity during the operation of wind and photovoltaic power plants causes practically no emissions of chemical compounds that are harmful to the environment. However, the production of their materials and components, as well as their post-use management after the end of their operation, is highly consumptive of energy and materials. For this reason, this article aims to assess the life cycle of a wind and photovoltaic power plant in the context of the sustainable development of energy systems. The objects of the research were two actual technical facilities—a 2 MW wind power plant and a 2 MW photovoltaic power plant, both located in Poland. The analysis of their life cycle was carried out on the basis of the LCA (life-cycle assessment) method, using the ReCiPe 2016 calculation procedure. The impact of the examined renewable energy systems was assessed under 22 impact categories and 3 areas of influence (i.e., human health, ecosystems, and resources), and an analysis was conducted for the results obtained as part of three compartments (i.e., air, water, and soil). The life cycle of the wind power plant was distinguished by a higher total potential negative environmental impact compared to the life cycle of the photovoltaic power plant. The highest levels of potential harmful impacts on the environment in both life cycles were recorded for areas of influence associated with negative impacts on human health. Emissions to the atmosphere accounted for over 90% of all emissions in the lifetimes of both the wind and the photovoltaic power plants. On the basis of the obtained results, guidelines were proposed for pro-ecological changes in the life cycle of materials and elements of the considered technical facilities for renewable energy sources, aimed at better implementation of the main assumptions of contemporary sustainable development (especially in the field of environmental protection).

## 1. Introduction

Year by year, the world needs much more energy—including for powering houses, industrial machines, and transport—due to the constantly growing population and ever-higher living standards. However, in order to counteract climate change, energy must increasingly come from more sustainable sources, with lower emissions of harmful substances into the environment. Thanks to advancements in knowledge, technology, and innovation, humanity is increasingly able to generate “cleaner” energy. Life and livelihoods, economies, and communities depend on a convenient, reliable, and affordable energy supply [1,2,3,4].

Most of the energy used today comes from crude oil and coal, which are non-renewable energy sources. By 2050, the world’s population is expected to grow to 9 billion (almost 2 billion more than today). Many people in developing economies will join the global middle class. They will buy various types of machinery and equipment that will consume significant amounts of energy. Global energy demand could double by the year 2050 compared to levels from the year 2000 [5,6,7].

Therefore, it is extremely important to counteract climate change caused by emissions of harmful substances and other destructive effects on the condition of the natural environment. In order to meet these challenges, radical changes in the global energy system and a number of new energy sources are needed. Fossil energy sources will continue to play an important role in the decades to come, but the use of alternative and innovative technologies will increasingly contribute to meeting the world’s growing energy needs and allow for more efficient, sustainable ways of using energy. These activities should be based primarily on cooperation, respect for the environment, and social responsibility [8,9,10,11].

Each source of energy has a certain effect on the environment. Renewable energy sources are considered to be the most environmentally friendly sources of energy—that is, those causing the least negative impact. Their exploitation is primarily aimed at slowing down climate change. They are a solution for global corporations, local entrepreneurs, and individual consumers. More and more countries are investing in alternative energy sources and supporting their development—for example, through subsidy programs or low-interest loans. Among the most popular renewable energy sources in the world are solar and wind energy installations. However, the life cycle of machines and devices, including those of renewable energy, is related to their specific demand for materials and energy [12,13,14,15].

Sustainable development is about finding solutions that guarantee further economic growth, which allow for the active inclusion of all social groups in development processes, while giving them the opportunity to benefit from this growth. Initially, it was understood as the need to reduce the negative impact of economies on the natural environment. Over the years, the concept has acquired a more complete understanding, aligning the essence of three development factors: respect for the environment, social progress, and economic growth. One of the most popular methods used in analyses in the area of sustainable development is LCA (life-cycle assessment. This method enables assessment of the potential environmental impact of both products and processes from the perspective of their entire life cycle (“from cradle to grave”)—starting from the extraction of raw materials, through production and exploitation, and ending with post-consumer management. Because of this, no stage of the product life cycle is skipped. As a result of the identification and quantitative assessment of the existing environmental loads, it is possible to analyze the potential impact of these loads on the environment and, consequently, to develop recommendations to reduce their negative impacts over their entire life cycle. LCA is a flexible method that allows for individual adjustment of the purpose and scope of the research of the object of analysis [16,17,18,19].

There are not many studies in the global literature in which analyses of the life cycles of wind and solar power plants have been performed using the relatively new method ReCiPe 2016. Most of the research conducted focuses only on the impact of the life cycle on GWP (global warming potential), ignoring other negative impacts of the systems under consideration, which reduce the quality of the environment, pose a threat to human health, and increase the depletion of raw materials; these also require detailed analyses, especially in view of the sustainable development of energy systems (see Section 4 for details). In Poland, unfortunately, analyses using the LCA methodology are still not very popular. This study tries to outline the local perspective of the environmental impact of selected renewable energy sources; hence, it was decided to study two real cases.

Therefore, the main objective of this study is to assess the life cycle of wind and photovoltaic power plants in the context of sustainable development of energy systems. It is based on a study of two real cases—a 2 MW wind power plant and a 2 MW photovoltaic power plant located in Poland.

## 2. Materials and Methods

### 2.1. Object and Plan of Analysis

The life-cycle assessment was carried out for an onshore 3-blade 2 MW horizontal wind power plant located in central Poland and a photovoltaic power plant with silicon monocrystalline photovoltaic panels (without a PV tracking system), with a capacity of 2 MW, located in the northern part of Poland. Life-cycle assessment of materials and elements of renewable energy systems is possible via the use of various models, including environmental LCA. This method was chosen as the model for assessing the potential impacts of wind and photovoltaic power plants on human health, ecosystems’ quality, and resource depletion. In accordance with the ISO 14040 (*environmental management*, *life-cycle assessment*, *principles and framework*) and ISO 14044 (*environmental management*, *life-cycle assessment*, *requirements and guidelines*) standards, the LCA analysis performed in this work included four stages: determination of goals and scope, life-cycle inventory (LCI), life-cycle impact assessment (LCIA), and interpretation (Figure 1) [20,21,22,23].

The research was started with the description of our goals and scope (details are provided in Section 2.2). Based on an earlier analysis of the state of the art and technology, it was found that the literature lacks a detailed life-cycle assessment of wind and photovoltaic power plants in the context of the sustainable development of energy systems. It was also extremely important when formulating the goals and scope to collect as many data as possible—and of the best possible quality—on the objects of analysis. This was possible thanks to cooperation with companies producing materials and elements of wind and photovoltaic power plants, which have a leading position in the European and domestic markets. A more detailed description of the second part of the research (LCI) is provided in Section 2.3. In the next step, a detailed analysis of the life cycle of the considered technical objects was carried out. The necessary simulation analyses were carried out using the SimaPro 9.3 software, using the ReCiPe 2016 calculation procedure. The course of this stage is presented in Section 2.4, and the obtained results and their interpretation as detailed in Section 3. The last part of the study (described in Section 2.5) included the interpretation of the obtained results and is presented in Section 3 and Section 4 [20,24,25,26,27].

### 2.2. Determination of Goals and Scope

The aim of the analysis carried out in this study was to compare the environmental impacts associated with the life cycle of wind and photovoltaic power plants (i.e., comparative analysis). The LCA analysis was used to determine whether there are differences in the magnitude of the environmental impacts generated during the life cycles of selected renewable energy sources operating based on two different technologies [20,28,29,30].

The systems of the analyzed technical objects were constructed in a comparable manner in terms of the depth and width of the analysis. The geographical scope was an area of Europe, as the companies that provided the data have a very strong position in the wider European market. The time ranges also covered the same range, since both wind power plants and the photovoltaic power plant have a life cycle of approximately 20 years. The cutoff level adopted for the research was 0.1%.

The conducted analysis can be classified as bottom-up and was mainly used to describe the existing reality (i.e., retrospective analysis), but also to model more pro-environmental solutions (i.e., prospective analysis). The level of advancement of the analysis classifies it among detailed analyses. The data used in the analysis were obtained from producers of the considered renewable energy systems or from SimaPro databases.

The functional unit was the value of the installed capacity in each of the research objects i.e., 2 MW. The environmental aspects of the assessment included 22 impact categories specific to the ReCiPe 2016 model (listed in Table 1). The obtained results were additionally grouped and compiled into three areas of influence: human health, ecosystems, and resources. Four areas of emission of individual chemical compounds were also specified: air, water, soil, and raw [20,31,32].

### 2.3. Life-Cycle Inventory (LCI)

Life-cycle inventory illustrates the system structure of a given technical object. All processes taking place in the life cycle of both wind and photovoltaic power plants relate to one another through material and energy streams. All collected data were assigned to unit processes and then validated based on the energy and mass balance. Models were systematically constructed and filled with data. The size of the inputs was equal to the size of the outputs. The inputs included main materials, support materials, and water requirements; the outputs were the main products and emissions. Information on key processes was obtained directly from the manufacturers of the materials and components. Data on less significant processes and materials from the point of view of environmental impact were obtained from databases included in the SimaPro 9.3 software (Ecoinvent 3.8 database). Due to the conclusion of a data confidentiality agreement with companies producing the analyzed renewable energy systems, all detailed information on the structure of the analysis objects and technological data are not disclosed in this study. [20,33,34,35,36].

The total mass of materials and elements of the tested wind power plant is about 2000 tons. The foundations have the largest share in the mass of the object—approximately 79% (of which approximately 96% is concrete, and the remaining 4% is steel). The other most important elements of the analyzed power plant include the tower, with a ~15% share in the weight of the entire facility (mostly made of steel); the rotor, with approximately 2% (about half of the mass of which is composed of blades made of polymers reinforced with fiberglass, while the other half is the hub, made mainly of nodular cast iron); and the nacelle, with a ~4% share (the elements of which are mainly made of cast iron (approximately 49% by weight of the nacelle), steel (approximately 38%), aluminum (approximately 4%), polymer materials (approximately 3%), and copper (approximately 2%)) (Figure 2 and Figure 3) (data obtained from manufacturers) (data obtained from the investor and producers).

The total mass of materials and elements of the tested photovoltaic power plant is around 300 tons. Single-crystalline silicon photovoltaic modules have the largest share in the mass of the object—approximately 62% (approximately 47% of which is solar glass, and approximately 45% of which is aluminum). The other most important elements of the analyzed power plant include the supporting structure, with a ~21% share in the weight of the entire facility (mostly made of steel); the inverter station, with a ~15% share (the elements of which are mainly made of steel (approximately 42%) and aluminum (approximately 38%)); and the electrical installation, with a ~2% share (mostly made of copper) (Figure 2 and Figure 3) (data obtained from the investor and producers).

### 2.4. Life-Cycle Impact Assessment (LCIA)

When determining the impact of the life cycle of a given technical facility on the environment, the third phase of the analysis—life-cycle impact assessment—is of key importance. Any methodological differences in the LCA approaches mainly relate to the LCIA phase, which is complex, with mandatory and optional elements. Mandatory elements include the selection of impact categories, category indicators, characterization models, classification, and characterization, while the optional elements include normalization, grouping, and weighting (Figure 4). The mandatory sequence of elements is strictly defined and must be preserved for parsing. The question of choice is whether and which optional elements will be used. As part of this research, all of the listed optional elements were used (i.e., normalization, grouping, and weighting). The analyses under this study were carried out using the SimaPro 9.3 software (PRé Sustainability, LE Amersfoort, Netherlands) with the Ecoinvent 3.8 database. The life-cycle assessment of wind and photovoltaic power plants was carried out using the ReCiPe 2016 method [20,37,38,39,40].

ReCiPe is one of the methods used in the life-cycle impact assessment (LCIA) stage. It was first developed in 2008 through cooperation between the Dutch National Institute for Public Health and the Environment (RIVM), Radboud University Nijmegen, Leiden University, and PRé Sustainability. The main purpose of the ReCiPe method is to convert life-cycle inventory results into indicator scores. Indicator scores express the potential magnitude of the impact on the environment. Under the ReCiPe method, indicators are determined on two levels—22 midpoint indicators (midpoint impact category) and 3 endpoint indicators (endpoint area of influence). Midpoint indicators focus on a single environmental problem, while endpoint indicators show the environmental impact on three higher aggregation levels (Figure 5) [41,42].

ReCiPe 2016 includes the broadest set of midpoint impact categories compared to other methods. Unlike other approaches (for example, Eco-indicator 99 or Impact 2002+), it does not include potential impacts from future extractions in the impact assessment but, rather, assumes that such impacts have been included in the inventory analysis. The ReCiPe 2016 method is an improvement of ReCiPe 2008 and previously used methods such as CML 2000 and Eco-indicator 99. In contrast with the previous version, ReCiPe 2016 also provides global characterization factors instead of only European ones [43,44].

Assigning LCI results to individual impact categories is referred to as classification. The use of appropriate, specialized software makes it possible to automate this stage. The SimaPro program was used for classification, which automatically assigns LCI results to individual impact categories based on a list of substances belonging to given calculation methods and databases included in the program [20,45,46].

Characterization and conversion of LCI results into the results of impact categories’ indicators are extremely complex processes. From a technical point of view, they come down to converting the LCI results through appropriate characterization parameters and showing them in the form of relative shares in each of the impact categories. The main calculation procedure used in this analysis was the ReCiPe 2016 method [20,47,48].

Normalization is understood as computing the magnitude of the results of a category indicator against reference information. It is used to determine the relative importance of the indicator results compared to a given region (e.g., Poland or Europe) or a person (for example, an average inhabitant of Poland or Europe) in a specific period of time. Normalization can also be used to prepare LCIA results for subsequent procedures, e.g., weighting. As part of the research, normalization was performed with the use of the SimaPro software. This was a necessary stage to carry out the next steps—grouping and weighting [20,49,50].

There are different evaluation methods and preferences for impact categories. Depending on the goal(s) and scope of the analysis, some may be more important than others. Additionally, they can be grouped, for instance, according to the emission level or scale (e.g., local, global). In the ReCiPe 2016 method, grouping takes place when the results of the impact categories’ indicators are summed up into three areas of influence and during the final aggregation to the total impact indicator [51,52].

The weighting process consists of determining and assigning a degree of importance (i.e., weighting factor) to individual impact categories, followed by multiplication by the normalized index results. Weighting should be performed on complete, internationally recognized sets of weighting factors that have been developed for all impact categories. Carrying out the weighting process allowed us to obtain the results in ecopoints (Pt). An ecopoint is a unit of measurement for the environmental impact of an individual, process, material, element, or product. The results presented in ecopoints reflect the influence of the average European on the environment. One thousand ecopoints is equal to the environmental impact of one European in one year. The more ecopoints a given unit, process, material, element, or product has, the greater its negative environmental impact. During the life-cycle analyses of wind and photovoltaic power plants, grouping and weighting were performed using the SimaPro program, and their results are presented in Section 3 [20,53,54].

### 2.5. Interpretation

On the one hand, the interpretation is the final part of the LCA analysis (i.e., the fourth phase); on the other hand, the interpretation process is still present for each of the three earlier stages of the procedure (i.e., determination of goal(s) and scope, LCI, and LCIA). The key purpose of the interpretation is the analysis of the results and their verification from the point of view of the previously established goal(s) and scope [20,55,56].

Completeness of the analysis was checked with a positive result. The data needed for the interpretation were complete. A compliance check was also carried out during the conducted research. The adopted assumptions, the methods used, the depth of the analysis, its detail, and the precision of data for both the materials and elements of the wind and photovoltaic power plants were consistent with the goals and scope of our research. The obtained results and their interpretation are presented in Section 3 and Section 4, respectively [57,58].

## 3. Results

The obtained results of this research are discussed in three sections. They present the grouping and weighting results in the unit ecopoints (Pt) (as discussed in Section 2.4). Section 3.1 compares the results of the potential environmental impact of the life cycle of wind and photovoltaic power plants in the 22 impact categories available under the ReCiPe 2016 method. In addition, the obtained results for five of the impact categories that potentially cause the greatest harmful impact on the environment are presented in detail. Section 3.2 presents the results of the analysis of the areas of influence and details the impact of one of them (also with the highest potential negative impact on the environment). Finally, Section 3.3 presents the results broken down into the three most important compartments (i.e., air, water, and soil).

### 3.1. Impact Categories

Among the 22 impact categories available under the ReCiPe 2016 method, the largest potentially negative impact on the environment, in the case of both the wind power plant (W) and photovoltaic power plant (PV) life cycle, stood out in the following impact categories: fine particulate matter formation (4.45 × 10^4^ Pt for the wind power plant and 1.59 × 10^4^ Pt for the photovoltaic power plant); global warming, human health (W—3.06 × 10^4^ Pt and PV—9.53 × 10^3^ Pt); human carcinogenic toxicity (W—5.82 × 10^3^ Pt and PV—1.49 × 10^3^ Pt); human non-carcinogenic toxicity (W—3.19 × 10^3^ Pt and PV—1.93 × 10^3^ Pt); and global warming, terrestrial ecosystems (W—3.06 × 10^3^ Pt and PV—9.55 × 10^2^ Pt) (Table 1). For this reason, these are discussed in more detail later in Section 3.1. Detailed results of the analyses for the remaining 17 impact categories are summarized in Appendix A.

In most of the analyzed impact categories, the impact of the wind power plant’s life cycle resulted in more potential negative environmental consequences compared to the life cycle of the photovoltaic power plant. Two impact categories were exceptions: ionizing radiation (PV—1.12 × 10^−1^ Pt and W—9.13 × 10^−2^ Pt) and marine ecotoxicity (PV—1.69 × 10^0^ Pt and W—6.27 × 10^−1^ Pt), most likely due to the specificity and high energy consumption of the photovoltaic modules’ production processes. In the production of photovoltaic modules, the most long-lasting and energy-consuming process is the cultivation of monocrystalline silicon crystals. The first step is the production of pure silicon from silicon dioxide by chemical methods. The material is then melted and crystallized by cooling. The individual crystals must not contain foreign atoms, so the process must take place under special conditions. For this purpose, a vacuum furnace is usually used, in which crystallization takes place without the access of gases—in particular, oxygen (this method was developed by J. Czochralski). After removal from the furnace, the crystals are mechanically processed in order to obtain the highest-quality silicon wafers, which are necessary for the production of photovoltaic modules. The life cycle of a wind power plant is characterized by a higher potential total negative impact on the environment (in total 8.98 × 10^4^ Pt) than the life cycle of a photovoltaic power plant (total 3.13 × 10^4^ Pt) (Table 1) [59].

As part of the impact category global warming, human health, the wind power plant’s life cycle resulted in greater emissions of chemical compounds that are hazardous to health (total impact at the level of 3.06 × 10^4^ Pt) when compared to the life cycle of the photovoltaic power plant (total 9.53 × 10^3^ Pt). In both cases, the largest share in the total emissions was carbon dioxide (W—2.73 × 10^4^ Pt and PV—8.05 × 10^3^ Pt), which was caused by the significantly high energy input from conventional sources during the production of the materials and components of both of the analyzed technical objects. The remaining greenhouse gases causing the most negative consequences in the life cycle of the wind power plant included methane (2.98 × 10^3^ Pt) and dinitrogen monoxide (1.98 × 10^2^ Pt), while those in the life cycle of the photovoltaic power plant also included methane (1.09 × 10^3^ Pt), as well as tetrafluoromethane, or CFC-14 (1.87 × 10^2^ Pt) (Table 2). Methane, which is also classified as a GHG, is mentioned much less frequently than carbon dioxide. However, it is a chemical compound that is dangerous to the environment. During the first 20 years in the atmosphere, the climatic effect of one ton of methane is about 85 times greater than that of one ton of carbon dioxide. The two largest sources of methane are the energy industry and agriculture. Methane from the energy industry is primarily emitted during the extraction, transport, and storage of fossil fuels such as coal, oil, and natural gas. Poland is the largest emitter of fossil methane in Europe. Methane, above all, strongly pollutes the air that people breathe. This can consequently lead to diseases such as asthma and emphysema. Current reports of the Intergovernmental Panel on Climate Change (IPCC) indicate that the global methane emissions should be reduced by approximately 50% in the next 20 years. Specific reduction steps have not been indicated, but the significant role of the energy sector in this regard is underlined. The key solution seems to be to abandon fossil fuels as quickly as possible and ensure that methane does not leak from closed mines or from abandoned gas or oil wells [60].

In the case of the impact category global warming, terrestrial ecosystems, the wind power plant’s life cycle also causes more negative environmental consequences in the considered scope (total impact equal to 3.06 × 10^3^ Pt) compared to the life cycle of the photovoltaic power plant (total 9.55 × 10^2^ Pt). Similarly to the previously discussed impact category, carbon dioxide (W—2.73 × 10^3^ Pt and PV—8.08 × 10^2^ Pt) has the greatest influence on the shaping of the impact of harmful emissions. As part of the life cycle of the wind power plant, significant emission values were also noted for methane (2.98 × 10^2^ Pt) and dinitrogen monoxide (1.98 × 10^1^ Pt), while in the life cycle of the photovoltaic power plant, tetrafluoromethane (CFC-14) (1.87 × 10^1^ Pt) and dinitrogen monoxide (9.38 × 10^0^ Pt) were significant (Table 3). Dinitrogen monoxide is one of the main greenhouse gases. Nitrogen oxides are among the most dangerous components of smog. Their toxicity is many times greater than that of carbon monoxide or sulfur dioxide. Dinitrogen monoxide, being the third most important long-term GHG, contributes significantly to global warming and is a substance that depletes stratospheric ozone significantly. Its greenhouse effect potential is approximately 140 times stronger than that of carbon dioxide. The average lifetime of this gas in the atmosphere is estimated to be over 100 years. Nitrogen oxides also have negative effects on human health. First of all, they irritate the respiratory system, posing a serious threat—especially to people suffering from asthma and lung diseases—and contributing to the exacerbation of ailments. Their negative impacts on health also occur indirectly, because nitrogen oxides contribute to soil acidification and the formation of carcinogenic compounds that penetrate into plants [61].

Another impact category with a key impact on the development of the level of harmful effects on the environment in the life cycle of both of the analyzed technical objects is fine particulate matter formation. Within this framework, more negative environmental consequences were noted for the life cycle of the wind power plant (jointly 4.45 × 10^4^ Pt) than for the photovoltaic power plant (jointly 1.59 × 10^4^ Pt). In the life cycle of both of the renewable energy installations mentioned above, three substances stood out with the highest levels of destructive emissions: particulates < 2.5 μm (W—2.58 × 10^4^ Pt and PV—6.43 × 10^3^ Pt), sulfur dioxide (W—1.35 × 10^4^ Pt and PV—7.15 × 10^3^ Pt), and nitrogen oxides (W—4.84 × 10^3^ Pt and PV—2.14 × 10^3^ Pt) (Table 4). Particulates < 2.5 μm are atmospheric aerosols no larger than 2.5 μm in diameter. Particles of this size are considered to be particularly dangerous to human health because they bypass many of the body’s defenses (such as nose hair and mucus) that act to trap particles before they penetrate deeper into the body. PM 2.5 particles can travel to the lungs, further into the alveoli and, eventually, into the bloodstream. These particles may contain toxic chemicals. This type of particulate matter can be responsible, inter alia, for the worsening of asthma, decreased lung function, cancer (of the lungs, throat, and larynx), abnormal heart rhythms, or inflammation of the blood vessels. The main sources of particulates < 2.5 μm are fossil fuel combustion, transport, and industry [61].

In the life cycles of both the wind power plant and the photovoltaic power plant, substances are produced that are sources of human carcinogenic toxicity. The higher emission levels of these substances were recorded for the first of the mentioned technical facilities (the total impact at the level of 5.82 × 10^3^ Pt). For both of the analyzed systems of renewable energy sources, chromium emissions to water had the greatest impact on the shaping of harmful environmental consequences in this impact category (W—3.83 × 10^3^ Pt and PV—1.23 × 10^3^ Pt) and the atmosphere (W—1.70 × 10^3^ Pt and PV—9.18 × 10^1^ Pt) (Table 5). Chromium is an element that occurs in two forms—Cr(III), which is found in food and is an important component of the diet; and Cr(VI), which is toxic to humans, and its derivatives are used in various industries. Chromium(III) is an element necessary for the proper functioning of the human body. It plays a very important role in the processes of insulin action and exerts a significant influence on glucose metabolism and its levels in the blood. It takes part in antioxidant processes and plays a role in the functioning of the immune system. It is part of several enzymes and is a catalyst for many chemical reactions. Both deficiency and excess of chromium can be harmful. The recommended, safe dietary intake of chromium is 50-200 μg per day. If this is exceeded, it has negative consequences for health. The reference values (norms) for the levels of chromium in the blood are 1.5–4.7 nmol/L. On the other hand, chromium(VI) is recognized as being carcinogenic, mutagenic, embryotoxic, and teratogenic. It is harmful to the digestive system, respiratory system, and skin. It damages the skin and mucous membranes, and inhaling chromium(VI) compounds can damage the nose, throat, lungs, stomach, and intestines. Chronic exposure to chromium(VI) is associated with negative effects on the immune, hematological, and reproductive systems as well as the risk of developing cancer. Chromium enters the body through food, inhalation, and through the skin. Chromic(VI) acid salts are used in many industries, e.g., in the metallurgical, chemical, and construction industries, and in the production of pigments, polymers, and glass products. For this reason, significant levels of emissions of this element are recorded in the life cycle of wind and photovoltaic power plants (especially during the production of their materials and components) [62].

The last of the impact categories, which is characterized by particularly high levels of negative environmental consequences in the life cycles of both of the considered technical objects, is human non-carcinogenic toxicity. Again, the wind power plant’s life cycle had a higher environmental impact in this area (total 3.19 × 10^3^ Pt) compared to the life cycle of the photovoltaic power plant (total 1.93 × 10^3^ Pt). The highest levels of emissions in the life cycle of both renewable energy systems were recorded for arsenic (air: W—9.46 × 10^2^ Pt and PV—4.88 × 10^2^ Pt, water: W—8.05 × 10^2^ Pt and PV—9.18 × 10^2^ Pt) and lead (W—7.45 × 10^2^ Pt and PV—2.83 × 10^2^ Pt) (Table 6). Arsenic is a substance that is harmful to human health. According to the World Health Organization (WHO), it is one of the top 10 chemical compounds of greatest importance to public health. Arsenic negatively affects the enzymatic processes in the cells of the body. It disrupts the work of the nervous, cardiovascular, respiratory, and reproductive systems, and has adverse effects on the production of hormones and the body’s immunity. Symptoms of arsenic poisoning usually occur after many years of exposure to this element (for example, due to its use in industry). Arsenic is not excreted from the body via metabolic processes—it is deposited in it, accumulates, and slowly poisons all systems and organs. Human activities contribute to the release of arsenic to the atmosphere, water, and soil. The sources of pollution include, among others, the mining and smelting of metal ores, and coal combustion. Arsenic is used, for example, in the production of semiconductors and to improve the quality of some metal alloys. This is the cause of significant emissions of this element in the life cycle of wind and photovoltaic power plants (mainly as part of the production processes of the materials and elements of these technical facilities) [63].

### 3.2. Areas of Influence

In the ReCiPe 2016 method, the results of the 22 impact categories indicators are summed up into three areas of influence—human health, ecosystems, and resources. The highest levels of harmful impact in the case of the wind power plant (jointly 8.44 × 10^4^ Pt), and the photovoltaic power plant (jointly 2.90 × 10^4^ Pt) were noted for areas of influence related to the impact on human health; in turn, the lowest were characterized by the impacts in the area of processes related to the depletion of resources. In the case of impacts on environmental quality, the life cycle of the wind power plant had more negative consequences compared to the photovoltaic power plant (Figure 6).

In the life cycle of both analyzed technical objects, the impact on human health accounted for approximately 94% of all negative consequences in relation to the environment (for ecosystems it was approximately 5%, and for resources it was approximately 1%). The triangle view also shows a higher level of harmful effects on the milieu generated throughout the life cycle of a wind power plant (blue color in the chart) compared to the life cycle of a photovoltaic power plant (red color in the chart) (Figure 7).

For this reason, the interactions occurring in the framework of human health are discussed in more detail later in Section 3.2. Detailed results of the analyses for the remaining two areas of influence are summarized in Appendix A.

Comparing the life cycle of wind and photovoltaic power plants, it is clear that the former has more negative effects (total: W—8.44 × 10^4^ Pt and PV—2.90 × 10^4^ Pt) in terms of the impact on human health. The substances causing the most destructive consequences in the discussed areas of influence in the life cycles of both of the considered technical renewable energy facilities included carbon dioxide (W—2.73 × 10^4^ Pt and PV—8.05 × 10^3^ Pt), particulates < 2.5 μm (W—2.58 × 10^4^ Pt and PV—6.43 × 10^3^ Pt), and sulfur dioxide (W—1.33 × 10^4^ Pt and PV—7.15 × 10^3^ Pt) (Table 7). Carbon dioxide occurs in the human body and is produced in it, playing an important role in maintaining the acid–base balance of the body, oxygen transport, and relaxation of smooth muscles in the walls of blood vessels, among other functions. It is part of the carbon cycle in nature and is a product of combustion and respiration; hence, the majority of it is emitted mainly in the processes of producing materials and elements of wind and photovoltaic power plants, which are characterized by significant energy and material consumption. Excess carbon dioxide in the atmosphere causes, among others, acidification of the water that absorbs it, which is important for many marine ecosystems. Above all, however, an increase in the concentration of this chemical compound enhances the greenhouse effect. This leads not only to an increase in the temperature of the Earth’s surface, but also to many other consequences. Increased concentrations of carbon dioxide in the inhaled air are among the most important factors that may increase the concentration of CO_2_ in the blood and cerebrospinal fluid. Its action causes dyspnea, hypercapnia, and subsequent cerebral edema. These cases seem to be rather extreme. However, carbon dioxide affects the human body every day, and most people feel the negative effects of excessive concentrations of this gas in the air. Increased concentrations of CO_2_ disrupt human cognitive processes (from making simple decisions to complex strategic thinking); its concentration achieved after several hours in a closed room has a negative impact on the effectiveness of learning, memory, and concentration. Carbon dioxide is a substance without which life on Earth and the functioning of organisms would not be possible, but the problem is not its existence itself, but the increase in its concentration, which is occurring at an increasingly faster pace [60,61].

### 3.3. Compartments

Among the most important types of emission of harmful substances to the environment, three can be distinguished—those to the atmospheric, water, and soil environments. In the case of both analyzed life cycles, emissions to the atmosphere cause the most negative environmental consequences (total: W—8.36 × 10^4^ Pt and PV—2.83 × 10^4^ Pt). For the lifetime of the wind power plant, emissions to the atmosphere account for over 94% of all emissions (water: approximately 6%, soil: <1%); on the other hand, for the life cycle of the photovoltaic power plant atmospheric emissions account for approximately 93% (water: approximately 7%, soil: <1%). Emissions to the atmospheric environment in the life cycles of both of the analyzed technical facilities cause the most negative consequences in the area of deteriorating human health (W—7.93 × 10^4^ Pt and PV—2.68 × 10^4^ Pt). For this reason, the human health area of influence is analyzed in more detail later in Section 3.3 (Figure 8).

The environmental impact of the wind power plant’s life cycle (total: 8.87 × 10^4^ Pt) in all compartments considered (i.e., air, water, and soil) was higher compared to the life cycle of photovoltaic power plants (total: 3.05 × 10^4^ Pt). In the case of emissions to the atmospheric environment, in the life cycles of both of the considered renewable energy systems, the most potential negative environmental consequences were recorded for the following impact categories: global warming, human health (W—3.06 × 10^4^ Pt and PV—9.53 × 10^3^ Pt); and fine particulate matter formation (W—4.45 × 10^4^ Pt and PV—1.59 × 10^4^ Pt). These are characterized in Section 3.1. As for the emissions to the water and soil environment, the impact categories causing the most potential harmful effects to the environment included human carcinogenic toxicity (water: W—3.88 × 10^3^ Pt and PV—1.28 × 10^3^ Pt, soil: W—1.28 × 10^1^ Pt and PV—1.36 × 10^1^ Pt) and human non-carcinogenic toxicity (water: W—8.95 × 10^2^ Pt and PV—8.88 × 10^2^ Pt, soil: W—7.12 × 10^1^ Pt and PV—3.30 × 10^0^ Pt). These impact categories are also discussed in more detail in Section 3.1 (Table 8).

## 4. Summary and Discussion

Global energy demand is growing, stimulated by the increase in the number and living standards of the population. Conventional energy uses huge amounts of coal, oil, gas, and other non-renewable fuels. Due to the processes of their acquisition, processing, and combustion, this sector of the economy is characterized by the highest share in the emission of pollutants into the environment. The use of renewable energy is always associated with some consumption of non-renewable resources, because the materials necessary to produce their materials and components are usually produced from fossil raw materials and with energy from conventional sources. Wind and photovoltaic power plants, however, are classified as “environmentally friendly” energy sources and meet the most important assumptions of contemporary sustainable development [64,65,66].

The largest share in the weight of wind power plants is contributed by the foundations—approximately 79% (including concrete (approximately 96%) and steel (4%)), followed by the tower—approximately 15% (mainly steel), nacelle—approximately 4% (including cast iron (approximately 49%), steel (approximately 38%), aluminum (approximately 4%), polymer materials (approximately 3%), and copper (approximately 2%)), and rotor—approximately 2% (including polymers reinforced with fiberglass (approximately 50%) and nodular cast iron (approximately 50%)). In turn, the largest share in the weight of the photovoltaic power plant is represented by photovoltaic modules—approximately 62% (including solar glass (approximately 47%) and aluminum (approximately 45%)), followed by the supporting structure—approximately 21% (mainly steel), inverter station—approximately 15% (including steel (approximately 42%) and aluminum (approximately 38%)), and electrical installation—approximately 2% (mainly copper). It is therefore visible that the analyzed technical objects of renewable energy are made of materials and elements that are largely characterized by high energy and material consumption in their production processes (Figure 2 and Figure 3).

The main goal of this study was achieved thanks to the assessment of the life cycles of a wind power plant and a photovoltaic power plant in the context of sustainable development of energy systems.

The life cycle of the wind power plant was distinguished by a higher potential total negative environmental impact compared to the life cycle of the photovoltaic power plant (Table 1). Within most impact categories, the impact of the wind power plant’s life cycle resulted in more harmful environmental impacts compared to the life cycle of the photovoltaic power plant.

In the case of the impact category global warming, human health, carbon dioxide had the largest share in the total emissions of harmful substances (which was conditioned by significant energy inputs from conventional sources during the production of the materials and elements of both analyzed technical objects) (Table 2). As in the previous impact category, carbon dioxide had the greatest impact on the levels of harmful emissions in the case of global warming, terrestrial ecosystems (Table 3). As part of the impact category fine particulate matter formation, in the life cycle of both considered renewable energy installations, the highest levels of hazardous emissions to the environment were characterized by three substances: particulates < 2.5 μm, and sulfur and nitrogen oxides (Table 4). Both for the wind power plant and the photovoltaic power plant, chromium emissions to water and the atmosphere had a key influence on the development of the total levels of harmful environmental effects in the area of impact category human carcinogenic toxicity (Table 5). On the other hand, in the case of the impact category human non-carcinogenic toxicity, the highest emission levels in the life cycles of both technical objects were recorded for arsenic and lead (Table 6).

The highest levels of potential negative environmental effects, in the case of both the wind power plant and the photovoltaic power plant, were recorded for areas of influence related to harmful effects on human health (Figure 6). In the life cycles of both considered renewable energy systems, the impact on human health accounted for approximately 94% of all negative impacts (Figure 7). The substances causing the most destructive consequences in this area included carbon dioxide, particulates < 2.5 μm, and sulfur dioxide (Table 7).

In the case of both analyzed life cycles, emissions to the atmosphere were the cause of the greatest number of negative environmental consequences (over 90% of all types of emissions). Substances entering the atmospheric environment cause the most harmful effects in the area of deteriorating human health (Figure 8). In the case of emissions to the atmosphere, in the life cycles of both the wind power plant and the photovoltaic power plant, the most potential negative environmental consequences were recorded for the impact categories global warming, human health and fine particulate matter formation (Table 8).

LCA analyses in the field of wind energy initially assumed power plants with a capacity of less than one MW as their research object. Schleisner [67] conducted one of the first studies of this type for a 500 kW turbine, while Ardente et al. [68] performed an analysis for a wind farm consisting of 11 turbines with a capacity of 660 kW each. However, several analyses were also carried out for wind energy systems with a high installed capacity, e.g., Alexandra et al. performed LCA tests for two onshore and two offshore wind power plants [69]. There have also been studies devoted to local issues, e.g., Martínez et al. [70] studied the impacts of the wind power plant life cycle on the environment in Spain; similar studies were conducted by Wagner et al. [71] in Germany, Schleisner [67] in Denmark, Ardente et al. [68] in Italy, Al-Behadili and El-Osta [72] in Libya, Kabir et al. [73] in Canada, Alsaleh et al. [74] in the United States, Vargas et al. [75] in Mexico, and Oebels et al. [76] in Brazil. In the case of this study, local conditions for Poland were taken into account. However, there are very few studies in the global literature in which analyses of the life cycle of wind power plants have been performed with the use of the relatively new method ReCiPe 2016. Most of the research conducted focuses only on the impact of the life cycle of the power plants on GWP (global warming potential), ignoring other negative impacts on the quality of the environment and human health and the depletion of raw materials, which also require detailed analyses—especially from the perspective of sustainable development of energy systems. Kabir et al. [73] examined three models of wind turbines of different power, discovering that the higher the power, the lower the CO_2_ emissions per kWh of produced energy. Oebels et al. [76] investigated a 1.5 MW power plant and found that the life cycle of its steel tower was mainly responsible for the highest greenhouse gas emissions. Chipindula et al. [77] performed LCA of offshore and onshore wind power plants with different installed capacity, obtaining results confirming that its increase translates into a reduction in carbon dioxide emissions per unit of electricity generated. Going further, Alsaleh et al. [74] analyzed a 2 MW turbine, considering different periods of operation of this type of facility, reaching the conclusion that the production stage causes the most GHG emissions to the atmosphere.

In the case of LCA analyses conducted for photovoltaic systems, a similar trend can be observed as for wind energy systems. The subject of prior research has usually been various types of materials from which PV modules are produced. The largest number of analyses was devoted to elements made of silicon, e.g., Alsema [78], Frankl et al. [79], Fthenakis and Kim [80], Dones and Frischknecht [81], and Kato et al. [82] studied the life cycle of modules made with single-crystalline silicon (sc-Si); on the other hand, the analyses of Alsema [78], Fthenakis and Alsema [83], Fthenakis and Kim [80], Dones and Frischknecht [81], Ito et al. [84,85], Kato et al. [82], Nomura et al. [86], and Oliver and Jackson [87] focused on multi-crystalline silicon (mc-Si), while the research conducted by Alsema [78], Ito et al. [85], and Kato et al. [82] included amorphous silicon (a-Si), and Bravi et al. [88] assessed modules made of multi-junction thin-film silicon (µc-Si). In the literature, works devoted to other materials can also be found, e.g., the studies of Fthenakis and Alsema [83], Fthenakis and Kim [80], and Ito et al. [85] were devoted to PV modules with cadmium telluride (CdTe), while Bravi et al. [88] analyzed modules made from copper indium gallium diselenide (CIGS), and Greijer et al. [89] investigated dye-sensitized solar cells (DSSCs). However, several studies were also carried out for high-power photovoltaic systems, e.g., Schaefer and Hagedorn [90] (2.5 MW; cells: sc-Si, mc-Si, and a-Si), Kato et al. [91] (10, 30, and 100 MW; cells: mc-Si and a-Si), and Kato et al. [92] (10, 30, and 100 MW; cells: CdTe). As in the case of wind energy power plants, for photovoltaic power plants one can also find studies devoted to local issues, e.g., Schaefer and Hagedorn [90] studied the environmental impact of the life cycle of PV systems in Germany; similar studies were carried out by Dones and Frischknecht [81] in Switzerland, Alsema [78] in the Netherlands, Bravi et al. [88] and Frankl et al. [79] in Italy, Fthenakis and Kim [80] in the United States, Kato et al. [82,91,92] and Nomura et al. [86] in Japan, and Ito at el. [84,85] in China. In the global literature on the assessment of the life cycle of photovoltaic power plants, there are very few studies in which the analyses were performed using the ReCiPe 2016 method. Most of the research conducted was focused on assessing the amounts of CO_2_ and other greenhouse gas emissions [78,79,80,81,82,83,84,85,86,87,88,89,90,91,92]. Other impacts lowering the quality of ecosystems, posing a threat to human health, and exacerbating the depletion of raw materials are usually not taken into account.

The increase in the share of renewable energy sources in the global energy balance contributes to a more economical use of fossil energy resources, improving the condition of the environment by reducing the emissions of pollutants to the atmosphere, water, soil, as well as the amounts of waste generated [93,94]. The use of energy generated by wind and photovoltaic power plants increases the level of energy security, creates new jobs, promotes regional development, and contributes to solving many environmental problems [73,74].

When designing individual stages of the life cycle of wind and photovoltaic power plants, manufacturers should pay special attention to the use of waste-free or—if this is not possible—low-waste technologies. If waste is generated, recovery and reuse should be increased. It is also important to maximize the levels of recovery and reuse of other substances and materials used in all technological processes. It is also necessary to limit the consumption of natural resources and pay attention to the use of substances with the lowest possible toxicity to the environment. Currently, it is also obvious that we should strive to increase the material and energy efficiency of the processes of producing materials and components, as well as to reduce the amount and range of emissions of the generated pollutants. All of this is related to the need to constantly follow technological and scientific development and to constantly analyze the possibility of launching procedures related to the implementation of *best available techniques* (BATs) [95,96,97,98,99,100].

Today, there is an urgent need to change the ways in which environmental resources are managed. Potentially, there are great opportunities to rationalize the use of nature’s resources. To achieve this, however, it is necessary to cultivate more sustainable attitudes not only in the area of production, but also in all other economic activities.

## Figures and Tables

**Figure 1 materials-15-07778-f001:**
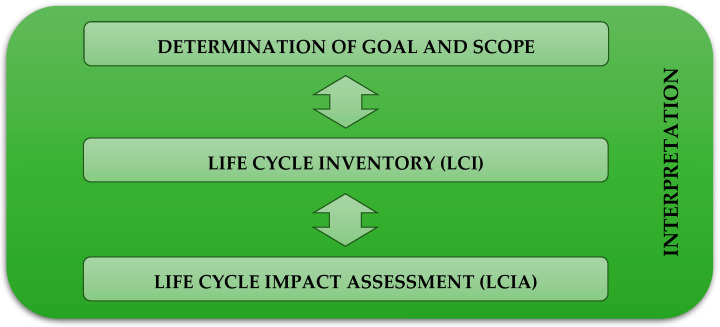
The stages of the LCA analysis (in accordance with the ISO 14040 and 14044 standards).

**Figure 2 materials-15-07778-f002:**
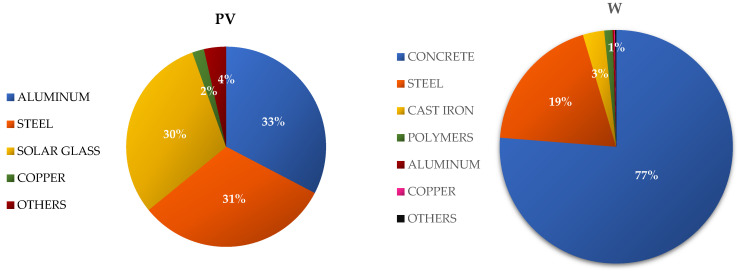
Percentage share of the most important materials from which the components of the wind and photovoltaic power plants were produced.

**Figure 3 materials-15-07778-f003:**
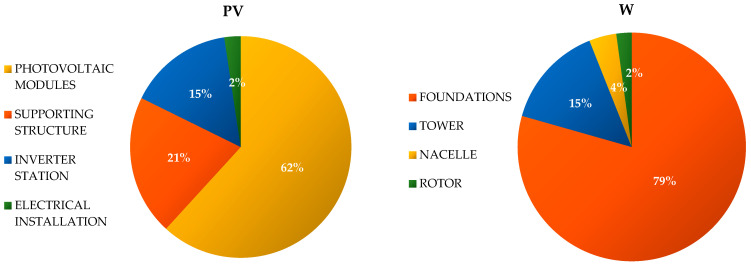
Percentage share of the most important components in the studied wind systems and photovoltaic power plants.

**Figure 4 materials-15-07778-f004:**
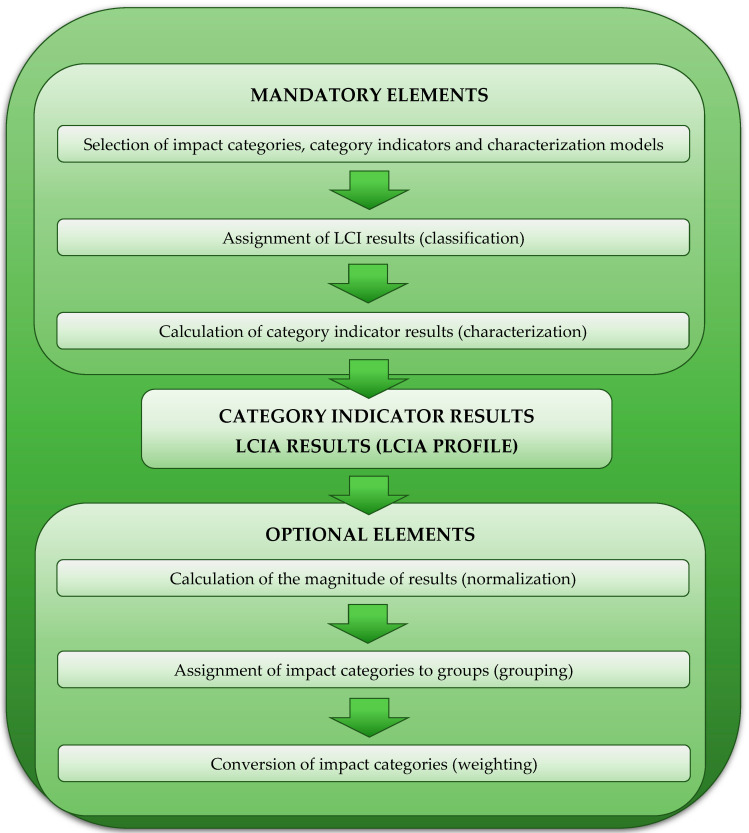
The elements of a life-cycle impact assessment (in accordance with the ISO 14040 and 14044 standards).

**Figure 5 materials-15-07778-f005:**
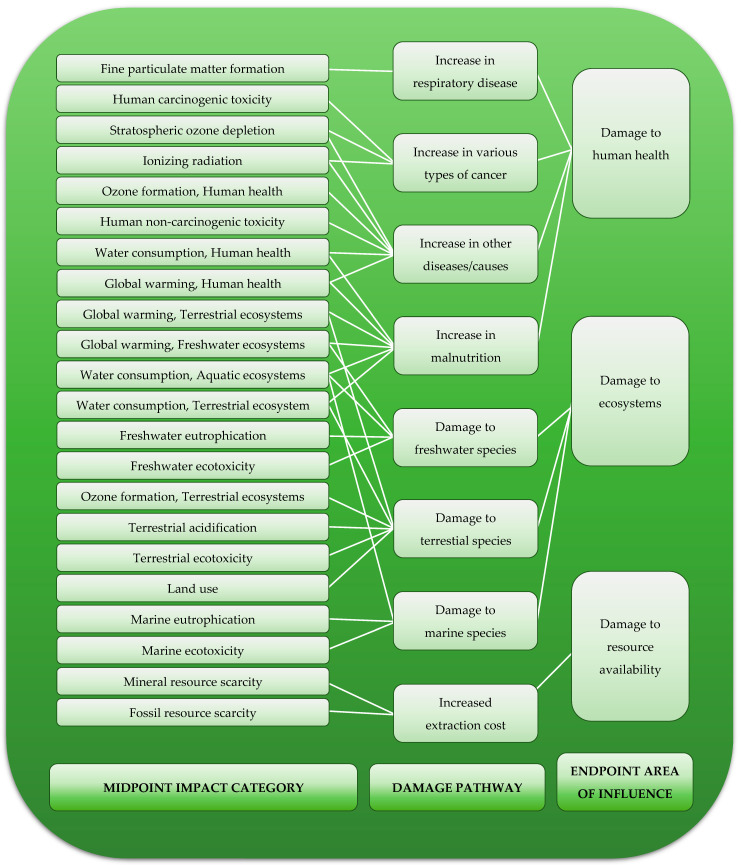
Overview of the impact categories that are covered in the ReCiPe 2016 method and their relation to the areas of influence.

**Figure 6 materials-15-07778-f006:**
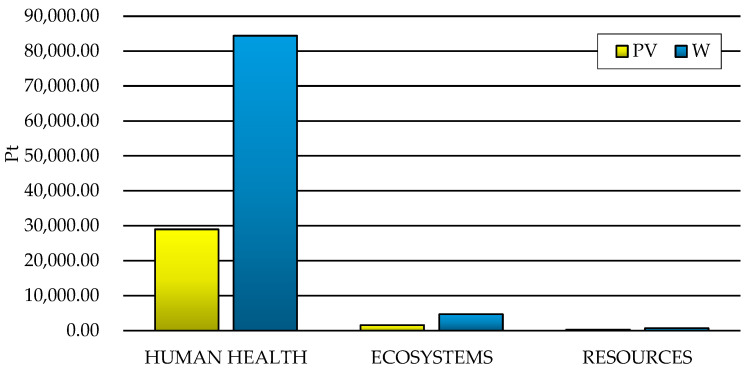
Grouping and weighting results of environmental effects for areas of influence present in the life cycle of wind (W) and photovoltaic (PV) power plants (unit: Pt).

**Figure 7 materials-15-07778-f007:**
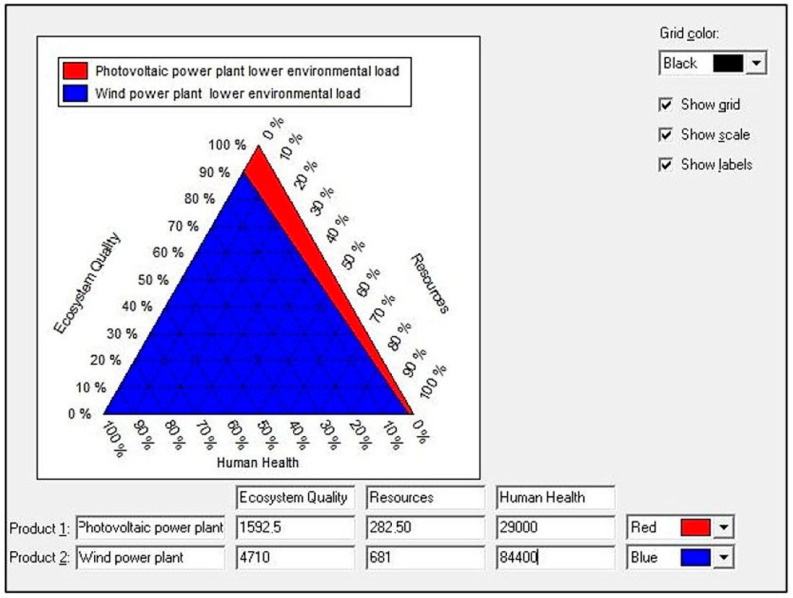
Grouping and weighting results of environmental effects for areas of influence present in the life cycle of wind (W) and photovoltaic (PV) power plants (unit: % and Pt)—triangle view.

**Figure 8 materials-15-07778-f008:**
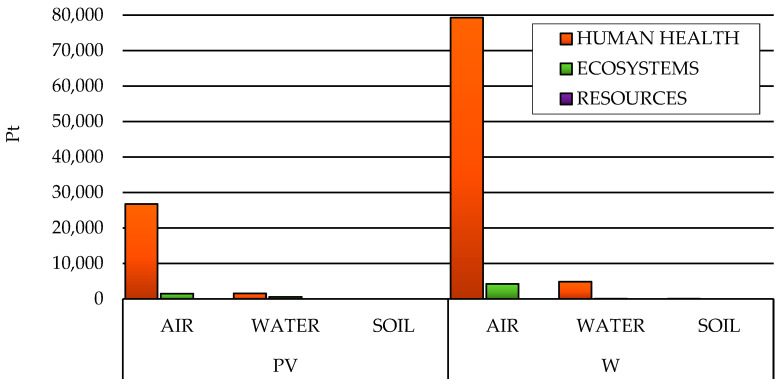
Grouping and weighting results of environmental effects for areas of influence, with consideration of compartments, present in the life cycle of wind (W) and photovoltaic (PV) power plants (unit: Pt).

**Table 1 materials-15-07778-t001:** Grouping and weighting results of environmental consequences occurring in the life cycle of wind (W) and photovoltaic (PV) power plants (unit: Pt).

No	Impact Category	PV	W
1.	Global warming, human health	9.53 × 10^3^	3.06 × 10^4^
2.	Global warming, terrestrial ecosystems	9.55 × 10^2^	3.06 × 10^3^
3.	Global warming, freshwater ecosystems	2.60 × 10^−2^	8.37 × 10^−2^
4.	Stratospheric ozone depletion	2.33 × 10^0^	5.06 × 10^0^
5.	Ionizing radiation	1.12 × 10^−1^	9.13 × 10^−2^
6.	Ozone formation, human health	2.93 × 10^1^	7.15 × 10^1^
7.	Fine particulate matter formation	1.59 × 10^4^	4.45 × 10^4^
8.	Ozone formation, terrestrial ecosystems	1.42 × 10^2^	3.58 × 10^2^
9.	Terrestrial acidification	3.73 × 10^2^	7.25 × 10^2^
10.	Freshwater eutrophication	1.12 × 10^1^	3.65 × 10^1^
11.	Marine eutrophication	4.83 × 10^−3^	6.73 × 10^−2^
12.	Terrestrial ecotoxicity	3.83 × 10^1^	1.16 × 10^2^
13.	Freshwater ecotoxicity	5.80 × 10^−2^	7.65 × 10^−1^
14.	Marine ecotoxicity	1.69 × 10^0^	6.27 × 10^−1^
15.	Human carcinogenic toxicity	1.49 × 10^3^	5.82 × 10^3^
16.	Human non-carcinogenic toxicity	1.93 × 10^3^	3.19 × 10^3^
17.	Land use	5.80 × 10^1^	3.39 × 10^2^
18.	Mineral resource scarcity	7.08 × 10^1^	1.14 × 10^2^
19.	Fossil resource scarcity	2.13 × 10^2^	5.68 × 10^2^
20.	Water consumption, human health	7.80 × 10^1^	2.31 × 10^2^
21.	Water consumption, terrestrial ecosystem	1.61 × 10^1^	6.65 × 10^1^
22.	Water consumption, aquatic ecosystems	2.33 × 10^−3^	9.36 × 10^−3^
**TOTAL**		**3.13 × 10^4^**	**8.98 × 10^4^**

Highlight—the highest levels of negative environmental consequences.

**Table 2 materials-15-07778-t002:** Grouping and weighting results of environmental effects for global warming, human health present in the life cycle of wind (W) and photovoltaic (PV) power plants (unit: Pt).

No	Substance	Compartment	PV	W
1.	Carbon dioxide, fossil	Air	8.05 × 10^3^	2.73 × 10^4^
2.	Carbon dioxide, land transformation	Air	2.73 × 10^1^	9.62 × 10^1^
3.	Dinitrogen monoxide	Air	9.38 × 10^1^	1.98 × 10^2^
4.	Ethane, hexafluoro-, HFC-116	Air	1.87 × 10^1^	3.65 × 10^0^
5.	Methane, biogenic	Air	1.74 × 10^1^	1.52 × 10^1^
6.	Methane, chlorodifluoro-, HCFC-22	Air	2.70 × 10^0^	2.31 × 10^1^
7.	Methane, fossil	Air	1.09 × 10^3^	2.98 × 10^3^
8.	Methane, tetrafluoro-, CFC-14	Air	1.87 × 10^2^	3.60 × 10^1^
9.	Sulfur hexafluoride	Air	3.38 × 10^1^	2.44 × 10^1^
10.	Remaining substances		5.75 × 10^−1^	3.62 × 10^0^
**TOTAL**			**9.53 × 10^3^**	**3.06 × 10^4^**

Highlight—the highest levels of negative environmental consequences.

**Table 3 materials-15-07778-t003:** Grouping and weighting results of environmental effects for global warming, terrestrial ecosystems present in the life cycle of wind (W) and photovoltaic (PV) power plants (unit: Pt).

No	Substance	Compartment	PV	W
1.	Carbon dioxide, fossil	Air	8.08 × 10^2^	2.73 × 10^3^
2.	Carbon dioxide, land transformation	Air	2.73 × 10^0^	9.63 × 10^0^
3.	Dinitrogen monoxide	Air	9.38 × 10^0^	1.98 × 10^1^
4.	Ethane, hexafluoro-, HFC-116	Air	1.87 × 10^0^	3.64 × 10^−1^
5.	Methane, biogenic	Air	1.73 × 10^0^	1.52 × 10^0^
6.	Methane, chlorodifluoro-, HCFC-22	Air	2.73 × 10^−1^	2.32 × 10^0^
7.	Methane, fossil	Air	1.09 × 10^0^	2.98 × 10^2^
8.	Methane, tetrafluoro-, CFC-14	Air	1.87 × 10^1^	3.61 × 10^0^
9.	Sulfur hexafluoride	Air	3.38 × 10^0^	2.44 × 10^0^
10.	Remaining substances		5.75 × 10^−2^	3.62 × 10^−1^
**TOTAL**			**9.55 × 10^2^**	**3.06 × 10^3^**

Highlight—the highest levels of negative environmental consequences.

**Table 4 materials-15-07778-t004:** Grouping and weighting results of environmental effects for fine particulate matter formation present in the life cycle of wind (W) and photovoltaic (PV) power plants (unit: Pt).

No	Substance	Compartment	PV	W
1.	Ammonia	Air	2.20 × 10^2^	2.69 × 10^2^
2.	Nitrogen oxides	Air	2.14 × 10^3^	4.84 × 10^3^
3.	Particulates, <2.5 μm	Air	6.43 × 10^3^	2.58 × 10^4^
4.	Sulfur dioxide	Air	7.15 × 10^3^	1.35 × 10^4^
5.	Sulfur oxides	Air	3.60 × 10^−4^	7.74 × 10^1^
6.	Remaining substances		9.50 × 10^−4^	4.94 × 10^−2^
**TOTAL**			**1.59 × 10^4^**	**4.45 × 10^4^**

Highlight—the highest levels of negative environmental consequences.

**Table 5 materials-15-07778-t005:** Grouping and weighting results of environmental effects for human carcinogenic toxicity present in the life cycle of wind (W) and photovoltaic (PV) power plants (unit: Pt).

No	Substance	Compartment	PV	W
1.	Arsenic	Air	2.95 × 10^1^	5.67 × 10^1^
2.	Arsenic	Water	5.25 × 10^1^	4.61 × 10^1^
3.	Benzene	Air	6.53 × 10^−1^	4.67 × 10^0^
4.	Benzo(a)pyrene	Air	1.58 × 10^0^	1.20 × 10^1^
5.	Cadmium	Air	1.10 × 10^0^	1.95 × 10^0^
6.	Chromium(VI)	Air	9.18 × 10^1^	1.70 × 10^3^
7.	Chromium(VI)	Water	1.23 × 10^3^	3.83 × 10^3^
8.	Chromium(VI)	Soil	1.36 × 10^1^	1.27 × 10^1^
9.	Dioxin, 2,3,7,8 Tetrachlorodibenzo-p-	Air	1.32 × 10^1^	2.20 × 10^1^
10.	Formaldehyde	Air	8.83 × 10^−1^	3.53 × 10^0^
11.	Lead	Air	3.43 × 10^0^	9.07 × 10^0^
12.	Nickel	Air	5.48 × 10^1^	1.10 × 10^2^
13.	Nickel	Water	3.23 × 10^−1^	1.40 × 10^0^
14.	PAH, polycyclic aromatic hydrocarbons	Air	2.63 × 10^0^	2.64 × 10^0^
15.	Remaining substances		2.75 × 10^−1^	1.28 × 10^0^
**TOTAL**			**1.49 × 10^3^**	**5.82 × 10^3^**

Highlight—the highest levels of negative environmental consequences.

**Table 6 materials-15-07778-t006:** Grouping and weighting results of environmental effects for human non-carcinogenic toxicity present in the life cycle of wind (W) and photovoltaic (PV) power plants (unit: Pt).

No	Substance	Compartment	PV	W
1.	Acephate	Soil	-	3.69 × 10^0^
2.	Acrolein	Air	-	2.18 × 10^0^
3.	Antimony	Air	3.33 × 10^0^	1.98 × 10^1^
4.	Antimony	Water	2.23 × 10^−1^	1.10 × 10^0^
5.	Arsenic	Air	4.88 × 10^2^	9.46 × 10^2^
6.	Arsenic	Water	9.18 × 10^2^	8.05 × 10^2^
7.	Barium	Air	9.48 × 10^−1^	1.94 × 10^0^
8.	Barium	Water	2.21 × 10^0^	3.93 × 10^0^
9.	Barium	Soil	6.18 × 10^−1^	1.47 × 10^0^
10.	Beryllium	Air	1.68 × 10^0^	3.86 × 10^0^
11.	Cadmium	Air	5.88 × 10^1^	9.80 × 10^1^
12.	Cadmium	Water	2.03 × 10^−1^	-
13.	Carbon disulfide	Air	1.59 × 10^2^	8.02 × 10^1^
14.	Chromium(VI)	Air	-	1.97 × 10^0^
15.	Chromium(VI)	Water	6.50 × 10^−1^	2.03 × 10^0^
16.	Copper	Air	4.30 × 10^−1^	1.05 × 10^0^
17.	Lead	Air	2.83 × 10^2^	7.45 × 10^2^
18.	Mercury	Air	-	6.84 × 10^0^
19.	Mercury	Water	-	1.21 × 10^0^
20.	Molybdenum	Air	2.11 × 10^−1^	1.24 × 10^0^
21.	Nickel	Air	7.23 × 10^−1^	1.45 × 10^0^
22.	Vanadium	Air	6.83 × 10^−1^	1.81 × 10^0^
23.	Vanadium	Water	1.95 × 10^−1^	-
24.	Zinc	Air	4.23 × 10^1^	3.15 × 10^2^
25.	Zinc	Water	3.38 × 10^−3^	8.09 × 10^1^
26.	Zinc	Soil	2.40 × 10^0^	6.55 × 10^1^
27.	Remaining substances		9.93 × 10^−1^	2.40 × 10^0^
**TOTAL**			**1.93 × 10^3^**	**3.19 × 10^3^**

Highlight—the highest levels of negative environmental consequences.

**Table 7 materials-15-07778-t007:** Grouping and weighting results of environmental effects for human health present in the life cycle of wind (W) and photovoltaic (PV) power plants (unit: Pt).

No	Substance	Compartment	PV	W
1.	Ammonia	Air	2.20 × 10^2^	2.69 × 10^2^
2.	Antimony	Air	3.33 × 10^0^	1.98 × 10^1^
3.	Arsenic	Air	5.18 × 10^2^	1.00 × 10^3^
4.	Arsenic	Water	9.70 × 10^2^	8.51 × 10^2^
5.	Benzo(a)pyrene	Air	-	1.20 × 10^1^
6.	Cadmium	Air	5.98 × 10^1^	1.00 × 10^2^
7.	Carbon dioxide, fossil	Air	8.05 × 10^3^	2.73 × 10^4^
8.	Carbon dioxide, land transformation	Air	2.73 × 10^1^	9.62 × 10^1^
9.	Carbon disulfide	Air	1.59 × 10^2^	8.02 × 10^1^
10.	Chromium(VI)	Air	9.18 × 10^1^	1.71 × 10^3^
11.	Chromium(VI)	Water	1.23 × 10^3^	3.84 × 10^3^
12.	Chromium(VI)	Soil	1.36 × 10^1^	1.27 × 10^1^
13.	Dinitrogen monoxide	Air	9.58 × 10^1^	2.02 × 10^2^
14.	Dioxin, 2,3,7,8 Tetrachlorodibenzo-p-	Air	1.32 × 10^1^	2.20 × 10^1^
15.	Ethane, hexafluoro-, HFC-116	Air	1.87 × 10^1^	-
16.	Lead	Air	2.85 × 10^2^	7.55 × 10^2^
17.	Methane, biogenic	Air	1.74 × 10^1^	1.52 × 10^1^
18.	Methane, chlorodifluoro-, HCFC-22	Air	-	2.34 × 10^1^
19.	Methane, fossil	Air	1.09 × 10^3^	2.98 × 10^3^
20.	Methane, tetrafluoro-, CFC-14	Air	1.87 × 10^2^	3.60 × 10^1^
21.	Nickel	Air	5.55 × 10^1^	1.11 × 10^2^
22.	Nitrogen oxides	Air	2.17 × 10^3^	4.91 × 10^3^
23.	Particulates, <2.5 μm	Air	6.43 × 10^3^	2.58 × 10^4^
24.	Sulfur dioxide	Air	7.15 × 10^3^	1.33 × 10^4^
25.	Sulfur hexafluoride	Air	3.38 × 10^1^	2.44 × 10^1^
26.	Sulfur oxides	Air	3.60 × 10^1^	7.74 × 10^1^
27.	Water (total)	Water	1.85 × 10^0^	2.95 × 10^1^
28.	Water, cooling, unspecified natural origin (total)	Raw	6.16 × 10^−1^	1.05 × 10^1^
29.	Water, lake (total)	Raw	4.34 × 10^−3^	7.00 × 10^−2^
30.	Water, river (total)	Raw	3.04 × 10^−2^	1.37 × 10^−1^
31.	Water, turbine use, unspecified natural origin (total)	Raw	2.02 × 10^2^	1.49 × 10^3^
32.	Water, unspecified natural origin (total)	Raw	2.37 × 10^−2^	7.70 × 10^−1^
33.	Water, well (total)	Raw	4.62 × 10^−2^	9.20 × 10^−2^
34.	Zinc	Air	6.03 × 10^−2^	3.15 × 10^2^
35.	Zinc	Water	4.22 × 10^−8^	8.09 × 10^1^
36.	Zinc	Soil	-	6.55 × 10^1^
37.	Remaining substances		3.95 × 10^−5^	1.07 × 10^2^
**TOTAL**			**2.90 × 10^4^**	**8.44 × 10^4^**

Highlight—the highest levels of negative environmental consequences.

**Table 8 materials-15-07778-t008:** Grouping and weighting results of environmental consequences for human health, with consideration of compartments, present in the life cycle of wind (W) and photovoltaic (PV) power plants (unit: Pt).

No	IMPACT CATEGORY	PV			W		
		AIR	WATER	SOIL	AIR	WATER	SOIL
1.	Global warming, human health	9.53 × 10^3^	-	-	3.06 × 10^4^	-	-
2.	Global warming, terrestrial ecosystems	9.55 × 10^2^	-	-	3.06 × 10^3^	-	-
3.	Global warming, freshwater ecosystems	2.60 × 10^−2^	-	-	8.37 × 10^−2^	-	-
4.	Stratospheric ozone depletion	2.33 × 10^0^	-	-	5.06 × 10^0^	-	-
5.	Ionizing radiation	1.09 × 10^−1^	2.83 × 10^−3^	-	7.75 × 10^−2^	1.38 × 10^−2^	-
6.	Ozone formation, human health	2.93 × 10^1^	-	-	7.15 × 10^1^	-	-
7.	Fine particulate matter formation	1.59 × 10^4^	-	-	4.45 × 10^4^	-	-
8.	Ozone formation, terrestrial ecosystems	1.42 × 10^2^	-	-	3.58 × 10^2^	-	-
9.	Terrestrial acidification	3.73 × 10^2^	-	-	7.25 × 10^2^	-	-
10.	Freshwater eutrophication	-	1.12 × 10^1^	7.08 × 10^−3^	-	3.64 × 10^1^	5.95 × 10^−2^
11.	Marine eutrophication	-	4.83 × 10^−3^	-	-	6.73 × 10^−2^	-
12.	Terrestrial ecotoxicity	3.83 × 10^1^	-	-	1.16 × 10^2^	-	-
13.	Freshwater ecotoxicity	1.39 × 10^−4^	4.95 × 10^−2^	2.63 × 10^−3^	5.85 × 10^−2^	6.89 × 10^−1^	1.75 × 10^−2^
14.	Marine ecotoxicity	1.58 × 10^−1^	1.53 × 10^0^	-	4.74 × 10^−1^	1.52 × 10^−1^	1.38 × 10^−3^
15.	Human carcinogenic toxicity	2.00 × 10^2^	1.28 × 10^3^	1.36 × 10^1^	1.93 × 10^3^	3.88 × 10^3^	1.28 × 10^1^
16.	Human non-carcinogenic toxicity	1.05 × 10^3^	8.88 × 10^2^	3.30 × 10^0^	2.22 × 10^3^	8.95 × 10^2^	7.12 × 10^1^
17.	Water consumption, human health	-	1.61 × 10^0^	-	-	1.60 × 10^2^	-
18.	Water consumption, terrestrial ecosystem	-	3.43 × 10^−1^	-	-	4.23 × 10^1^	-
19.	Water consumption, aquatic ecosystems	-	3.08 × 10^−4^	-	-	6.18 × 10^−3^	-
**TOTAL**		**2.82 × 10^4^**	**2.18 × 10^3^**	**1.69 × 10^1^**	**8.36 × 10^4^**	**5.01 × 10^3^**	**8.41 × 10^1^**

Highlight—the highest levels of negative environmental consequences.

## Data Availability

The data used in this study are available upon request from the corresponding author. The data are not publicly available due to privacy policies.

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
