# Peer review of "Assessment of the Life Cycle of a Wind and Photovoltaic Power Plant in the Context of Sustainable Development of Energy Systems"

_materials, 2022, doi:10.3390/ma15217778_

Round 1

Reviewer 1 Report

The authors of this manuscript present a research article about assessment of the life cycle of a wind and photovoltaic power plant for sustainable development of energy systems.

The methods used are adequate and described well in detail. However, there are some issues with the manuscript that should be addressed before the manuscript could be published. The manuscript mainly focusses on the comparison of emissions produced in the life cycle of the wind and photovoltaic power plants that manifest their environmental friendly nature not sustainablity. In this regard, the title of the manuscript is a bit misleading and should match what is described in the manuscript.

Major Points:

·       In introduction, the life cycle assessment (LCA) method is mention that is used for sustainability but no solid reason is given besides its ‘popularity’.

·       In section 3.1, the exceptional impact categories for ionizing radiation (PV – 1.12.10-1 Pt and W – 9.13.10-2 Pt) and marine ecotoxicity (PV –1.69.100 Pt and W – 6.27.10-1Pt), what are the energy consumption processes due to which this exceptional case is observed? It needs a better description.

·       In section 3.1, while describing the chromium as a source of human carcinogenity, the borderline value is not given which distinguishes the deficiency and excees level, using the sentence “too much chromium” is not the scientific wording. “Too much” must be defined in terms of mass of Cr in the body. Also Cr behaves differently in different oxidation states so in what oxidation state it is toxic?

 Minor Points:

·       In the line No. 89 the word “Poland” is used 2 times. Remove the duplicate.

·       In the line No.92 the word “na” is wrong, use the proper word to complete the sentence.

·       There are certain typographical mistakes as well in the manuscript that must be checked thoroughly and corrected throughout the manuscript.

Author Response

RESPONSE TO REVIEWER 1 COMMENTS

Dear Reviewer,

Thank you for taking the time to read our article and for the review. We appreciate all comments and suggestions as they will help us improve our research workflow in the future.

 Detailed answers to all comments are provided below. Amendments in the text of the article are marked in red.

  1. The authors of this manuscript present a research article about assessment of the life cycle of a wind and photovoltaic power plant for sustainable development of energy systems. The methods used are adequate and described well in detail. However, there are some issues with the manuscript that should be addressed before the manuscript could be published. The manuscript mainly focusses on the comparison of emissions produced in the life cycle of the wind and photovoltaic power plants that manifest their environmental friendly nature not sustainablity. In this regard, the title of the manuscript is a bit misleading and should match what is described in the manuscript.

Thank you very much for this attention.

Sustainable development includes environmental, economic and social aspects. In the article, particular attention is paid to the aspect of the potential impact on the environment, which, however, also has an impact on the economy and society.

The summary briefly presents the most important reasons for undertaking research in the field of assessing the life cycle of wind and photovoltaic power plant:

Lines 16-23: The conversion of kinetic energy of wind and solar radiation into electricity during the operation of wind and photovoltaic power plants causes practically no emissions of chemical compounds harmful to the environment. However, the production of their materials and components, and then their post-use management after the end of their operation, are very energy and material consuming. For this reason, this article aims to assess the lifecycle of a wind and photovoltaic power plant in the context of the sustainable development of energy systems. The object of the research were two actual technical facilities - 2 MW wind power plant and 2 MW photovoltaic power plant, located in Poland.

In the introduction, an attempt was made to discuss the above-mentioned causes, referring to the problem of increasing demand for energy from more sustainable sources and the related climate changes, caused by a high share of conventional energy sources in the energy mixes of most countries of the world:

Lines 40-46: Year by year, the world needs much more energy, including powering houses, industrial machines or transport, due to the constantly growing population and ever higher living standards. However, in order to counteract climate change, energy must increasingly come from more sustainable sources, with lower emissions of harmful substances into the environment. Thanks to the knowledge, technology and innovation, man is able to generate more "cleaner" energy. Life and livelihoods, economies and communities depend on a convenient, reliable and affordable energy supply [1–4].

Lines 47-52: Most of the energy used today comes from crude oil and coal, which are non-renewable energy sources. By 2050, the world's population is expected to grow to 9 billion (almost 2 billion more than today). Many people in developing economies will join the global middle class. They will buy various types of machinery and equipment that will consume significant amounts of energy. Global energy demand could double by year 2050 compared to levels from year 2000 [5–7].

The need to introduce changes to the energy system was also mentioned in order to reduce the emission of harmful compounds to the environment and make it more sustainable:

Lines 53-61: Therefore, it is extremely important to counteract climate change caused by emissions of harmful substances and other destructive effects on the condition of the natural environment. In order to meet these challenges, radical changes in the global energy system and a number of new energy sources are needed. Fossil energy sources will continue to play an important role in the decades to come, but the use of alternative and innovative technologies will increasingly contribute to meeting the world's growing energy needs and allow for more efficient, sustainable ways of using energy. These activities should be based primarily on cooperation, respect for the environment and social responsibility [8–11].

It was also noted that each energy source, also renewable, has a certain impact on the environment. Renewable energy sources are considered the most "friendly" to the environment, but their life cycle is also associated with a specific demand for energy and matter, as well as some negative environmental consequences. Therefore, it was considered necessary to look more closely at the environmental impacts occurring in their life cycle in order to limit the harmful effects and to make efforts to make them more sustainable:

Lines 62-71: Each source of energy has a certain effect on the environment. Renewable energy sources are considered to be the most environmentally-friendly sources of energy, those causing the least negative impact that is. Their exploitation is primarily aimed at slowing down climate change. They are a solution for global corporations, local entrepreneurs and individual consumers. More and more countries are investing in alternative energy sources and supporting their development, for example through subsidy programs or low-interest loans. One of the most popular renewable energy sources in the world are solar and wind energy installations. However, the life cycle of machines and devices, including those of renewable energy, is related to a specific demand for matter and energy [12–15].

In addition, it was decided to more precisely justify the choice of the LCA method as a research tool:

Lines 72-88: Sustainable development is about finding solutions that guarantee further economic growth, which allow for the active inclusion of all social groups in development processes, while giving them the opportunity to benefit from this growth. Initially, it was understood as the need to reduce the negative impact of economies on the natural environment. Over the years, the concept has acquired a more complete understanding, aligning the essence of the three development factors: respect for the environment, social progress and economic growth. One of the most popular methods used in analyzes in the area of sustainable development is LCA - Life Cycle Assessment. This method allows to assess the potential impact on the environment of both products and processes in the perspective of their entire life cycle ('from cradle to grave') – starting from the extraction of raw materials, through production, exploitation, ending with post-consumer management. Because of this, no stage of the product life cycle is skipped. As a result of the identification and quantitative assessment of the existing environmental loads, it is possible to analyze the potential impact of these loads on the environment, and consequently – developing recommendations to reduce their negative impacts over the entire life cycle. LCA is a flexible method that allows for individual adjustment of the purpose and scope of the research of the analyzed object of analysis [16–19].

A broader discussion of the authors' motivations to undertake the analyzes carried out as part of this article has also been added:

Lines 89-98: There are not many studies in the world literature in which analyzes of the life cycle of wind and photovoltaic power plants would be performed using a relatively new method ReCiPe 2016. Most of the research conducted focuses only on the impact of the life cycle on GWP (Global Warming Potential), ignoring other negative impacts of the systems under consideration, which reduce the quality of the environment, pose a threat to human health and increase the depletion of raw materials, which also require detailed analyzes, especially in view of the sustainable development of energy systems (see section 5 for details). In Poland, unfortunately, analyzes using the LCA methodology are still not very popular. The study tries to outline the local perspective of the environmental impact of selected renewable energy sources, hence it was decided to study two real cases.

We apologize for the inconvenience and hope that we have managed to explain the motivation for accepting the title of the study : „Assessment of the life cycle of a wind and photovoltaic power plant in the context of sustainable development of energy systems”.

  1. In introduction, the life cycle assessment (LCA) method is mention that is used for sustainability but no solid reason is given besides its ‘popularity’.

Thank you very much for this valuable remark.

As mentioned before, the introduction has been supplemented with the justification for choosing LCA as the method used for the research:

Lines 79-88: Sustainable development is about finding solutions that guarantee further economic growth, which allow for the active inclusion of all social groups in development processes, while giving them the opportunity to benefit from this growth. Initially, it was understood as the need to reduce the negative impact of economies on the natural environment. Over the years, the concept has acquired a more complete understanding, aligning the essence of the three development factors: respect for the environment, social progress and economic growth. One of the most popular methods used in analyzes in the area of sustainable development is LCA - Life Cycle Assessment. This method allows to assess the potential impact on the environment of both products and processes in the perspective of their entire life cycle ('from cradle to grave') – starting from the extraction of raw materials, through production, exploitation, ending with post-consumer management. Because of this, no stage of the product life cycle is skipped. As a result of the identification and quantitative assessment of the existing environmental loads, it is possible to analyze the potential impact of these loads on the environment, and consequently – developing recommendations to reduce their negative impacts over the entire life cycle. LCA is a flexible method that allows for individual adjustment of the purpose and scope of the research of the analyzed object of analysis [16–19].

  1. In section 3.1, the exceptional impact categories for ionizing radiation (PV – 1.12.10-1 Pt and W – 9.13.10-2 Pt) and marine ecotoxicity (PV –1.69.100 Pt and W – 6.27.10-1Pt), what are the energy consumption processes due to which this exceptional case is observed? It needs a better description.

Thank you very much for the above question.

In section 3.1, the description of the production process of photovoltaic modules was extended, in which the production of silicon monocrystals by the Czochralski method is characterized by the highest energy and material consumption:

Lines 305-321: In most of the analyzed impact categories, the impact of the wind power plant life cycle resulted in more potential negative environmental consequences compared to the life cycle of the photovoltaic power plant. Two were the exceptions impact categories: ionizing radiation (PV – 1.12×10-1 Pt and W – 9.13×10-2 Pt) and marine ecotoxicity (PV – 1.69×100 Pt and W – 6.27×10-1 Pt), which is most likely due to the specificity and high energy consumption of the photovoltaic modules production processes. In the production of photovoltaic modules, the most long-lasting and energy-consuming process is the cultivation of monocrystalline silicon crystals. The first step is the production of pure silicon from silicon dioxide by chemical methods. The material is then melted and then crystallized by cooling. The single crystal must not contain foreign atoms, so the process must take place under special conditions. For this purpose, a vacuum furnace is usually used, in which crystallization takes place without the access of gases, in particular oxygen (this method was developed by J. Czochralski). After removing from the furnace, the crystals are mechanically processed in order to obtain the highest quality silicon wafers, necessary for the production of photovoltaic modules. The life cycle of a wind power plant is characterized by a higher potential total negative impact on the environment (in total 8.98×104 Pt) than the life cycle of a photovoltaic power plant (total 3.13×104 Pt) (Table 1).

  1. In section 3.1, while describing the chromium as a source of human carcinogenity, the borderline value is not given which distinguishes the deficiency and excees level, using the sentence “too much chromium” is not the scientific wording. “Too much” must be defined in terms of mass of Cr in the body. Also Cr behaves differently in different oxidation states so in what oxidation state it is toxic?

Thank you very much for the above question.

Information contained in lines 395-422 have been supplemented with the recommended values of the daily intake of chromium and the norms of the content of this element in the blood. It also clarified which oxidation state chromium exhibits toxic properties and on which it is an important part of the diet:

In both the life cycle of the wind power plant and the photovoltaic power plant, substances that are produced are the source of human carcinogenic toxicity. Their higher emission level was recorded for the first of mentioned technical facilities (the total impact at the level of 5.82×103 Pt). For both analyzed systems of renewable energy sources, chromium emissions to water had the greatest impact on the shaping of harmful environmental consequences in this impact category (W – 3.83×103 Pt and PV – 1.23×103 Pt) and atmosphere (W – 1.70×103 Pt and PV – 9.18×101 Pt) (Table 5). Chromium is an element that occurs in two forms - Cr (III) found in food, which is an important component of the diet, and Cr (VI), which is toxic to humans, and its derivatives are used in various industries. Chromium III is an element necessary for the proper functioning of the human body.. It plays a very important role in the processes of insulin action and exerts a significant influence on glucose metabolism and its level in the blood. It takes part in antioxidant processes and plays a role in the functioning of the immune system. It is part of enzymes and is a catalyst for many chemical reactions. Both chromium deficiency and excess can be harmful. The recommended, safe dietary intake of chromium is 50-200 μg per day. If it is exceeded, it has negative consequences for health. The reference values (norms) for the level of chromium in the blood are 1.5-4.7 nmol/l. On the other hand, chromium VI is recognized as being carcinogenic, mutagenic, embryotoxic and teratogenic. It is harmful to the digestive system, respiratory system and skin. It damages the skin and mucous membranes, and inhaling chromium VI compounds can damage the nose, throat, lungs, stomach and intestines. Chronic exposure to chromium VI is associated with its negative effects on the immune, hematological and reproductive systems and the risk of cancer development. Chromium reaches the body through food, inhalation and through the skin. Chromic VI acid salts are used in many industries - in the metallurgical, chemical and construction industries, in the production of pigments, polymers and glass products. For this reason, a significant level of emissions of this element is recorded in the lifecycle of winds and photovoltaic power plants (especially during the production of their materials, materials and components) [62].

  1. In the line No. 89 the word “Poland” is used 2 times. Remove the duplicate.

Thank you for your attention.

Duplicate of word “Poland” is removed from line no. 89 (line 108 in new version of article).

  1. In the line No.92 the word “na” is wrong, use the proper word to complete the sentence.

Thank you very much for this valuable remark.

Changed wording from “na” on “on”:

Lines 110-112: This method has been chosen as the model of assessing the potential impact wind and photovoltaic power plants on human health, ecosystems quality and resource depletion.

  1. There are certain typographical mistakes as well in the manuscript that must be checked thoroughly and corrected throughout the manuscript.

Thank you very much for attention.

A detailed review has been made and all typographical errors noted in the text have been corrected.

Reviewer 2 Report

This paper is not ready for publication now, which still needs further revision and following comments can be referred.

1.      The overall structure of this paper should be organized in a more logic manner, while current version is confusing regarding to the contributions of this paper.

2. The motivation is unclear.

3. The novelty of this paper is limited

4. The conclusion is not conclusive. The authors just repeat the method and experimental results without drawing any conclusion.

5. The first three terms in points are almost the same thing.

6. The manuscript needs careful proofreading to further improve the English language.

7.      Other details should be checked again, including but not limited to the spelling, mathematical expressions and grammar.

Author Response

RESPONSE TO REVIEWER 2 COMMENTS

Dear Reviewer,

Thank you for taking the time to read our article and for the review. We appreciate all comments and suggestions as they will help us improve our research workflow in the future.

 Detailed answers to all comments are provided below. Amendments in the text of the article are marked in red.

  1. The overall structure of this paper should be organized in a more logic manner, while current version is confusing regarding to the contributions of this paper.

Thank you for the note that puts the article in order.

Efforts were made to change the structure of the article to be more accessible and logical. The introduced changes are marked in red in the new version of the article.

  1. The motivation is unclear.

Thank you very much for this valuable remark.

The summary briefly presents the most important reasons for undertaking research in the field of assessing the life cycle of wind and photovoltaic power plant:

Lines 16-23: The conversion of kinetic energy of wind and solar radiation into electricity during the operation of wind and photovoltaic power plants causes practically no emissions of chemical compounds harmful to the environment. However, the production of their materials and components, and then their post-use management after the end of their operation, are very energy and material consuming. For this reason, this article aims to assess the lifecycle of a wind and photovoltaic power plant in the context of the sustainable development of energy systems. The object of the research were two actual technical facilities - 2 MW wind power plant and 2 MW photovoltaic power plant, located in Poland.

In the introduction, an attempt was made to discuss the above-mentioned causes, referring to the problem of increasing demand for energy from more sustainable sources and the related climate changes, caused by a high share of conventional energy sources in the energy mixes of most countries of the world:

Lines 40-46: Year by year, the world needs much more energy, including powering houses, industrial machines or transport, due to the constantly growing population and ever higher living standards. However, in order to counteract climate change, energy must increasingly come from more sustainable sources, with lower emissions of harmful substances into the environment. Thanks to the knowledge, technology and innovation, man is able to generate more "cleaner" energy. Life and livelihoods, economies and communities depend on a convenient, reliable and affordable energy supply [1–4].

Lines 47-52: Most of the energy used today comes from crude oil and coal, which are non-renewable energy sources. By 2050, the world's population is expected to grow to 9 billion (almost 2 billion more than today). Many people in developing economies will join the global middle class. They will buy various types of machinery and equipment that will consume significant amounts of energy. Global energy demand could double by year 2050 compared to levels from year 2000 [5–7].

The need to introduce changes to the energy system was also mentioned in order to reduce the emission of harmful compounds to the environment and make it more sustainable:

Lines 53-61: Therefore, it is extremely important to counteract climate change caused by emissions of harmful substances and other destructive effects on the condition of the natural environment. In order to meet these challenges, radical changes in the global energy system and a number of new energy sources are needed. Fossil energy sources will continue to play an important role in the decades to come, but the use of alternative and innovative technologies will increasingly contribute to meeting the world's growing energy needs and allow for more efficient, sustainable ways of using energy. These activities should be based primarily on cooperation, respect for the environment and social responsibility [8–11].

It was also noted that each energy source, also renewable, has a certain impact on the environment. Renewable energy sources are considered the most "friendly" to the environment, but their life cycle is also associated with a specific demand for energy and matter, as well as some negative environmental consequences. Therefore, it was considered necessary to look more closely at the environmental impacts occurring in their life cycle in order to limit the harmful effects and to make efforts to make them more sustainable:

Lines 62-71: Each source of energy has a certain effect on the environment. Renewable energy sources are considered to be the most environmentally-friendly sources of energy, those causing the least negative impact that is. Their exploitation is primarily aimed at slowing down climate change. They are a solution for global corporations, local entrepreneurs and individual consumers. More and more countries are investing in alternative energy sources and supporting their development, for example through subsidy programs or low-interest loans. One of the most popular renewable energy sources in the world are solar and wind energy installations. However, the life cycle of machines and devices, including those of renewable energy, is related to a specific demand for matter and energy [12–15].

In addition, it was decided to more precisely justify the choice of the LCA method as a research tool:

Lines 72-88: Sustainable development is about finding solutions that guarantee further economic growth, which allow for the active inclusion of all social groups in development processes, while giving them the opportunity to benefit from this growth. Initially, it was understood as the need to reduce the negative impact of economies on the natural environment. Over the years, the concept has acquired a more complete understanding, aligning the essence of the three development factors: respect for the environment, social progress and economic growth. One of the most popular methods used in analyzes in the area of sustainable development is LCA - Life Cycle Assessment. This method allows to assess the potential impact on the environment of both products and processes in the perspective of their entire life cycle ('from cradle to grave') – starting from the extraction of raw materials, through production, exploitation, ending with post-consumer management. Because of this, no stage of the product life cycle is skipped. As a result of the identification and quantitative assessment of the existing environmental loads, it is possible to analyze the potential impact of these loads on the environment, and consequently – developing recommendations to reduce their negative impacts over the entire life cycle. LCA is a flexible method that allows for individual adjustment of the purpose and scope of the research of the analyzed object of analysis [16–19].

A broader discussion of the authors' motivations to undertake the analyzes carried out as part of this article has also been added:

Lines 89-98: There are not many studies in the world literature in which analyzes of the life cycle of wind and photovoltaic power plants would be performed using a relatively new method ReCiPe 2016. Most of the research conducted focuses only on the impact of the life cycle on GWP (Global Warming Potential), ignoring other negative impacts of the systems under consideration, which reduce the quality of the environment, pose a threat to human health and increase the depletion of raw materials, which also require detailed analyzes, especially in view of the sustainable development of energy systems (see section 5 for details). In Poland, unfortunately, analyzes using the LCA methodology are still not very popular. The study tries to outline the local perspective of the environmental impact of selected renewable energy sources, hence it was decided to study two real cases.

We apologize for the inconvenience, and we hope that we have managed to explain the motivation for undertaking research in the field of life cycle assessment of a selected wind and photovoltaic power plant.

  1. The novelty of this paper is limited.

Thank you for your attention.

There are research in the life cycle of wind and photovoltaic power plants in the world literature. However, in Poland, analyzes using the LCA methodology are still not very popular. The study tried to outline the local perspective of the environmental impact of selected renewable energy sources, hence it was decided to carry out a study of two real cases - a 2 MW wind power plant and a 2 MW photovoltaic power plant located in Poland. By maintaining a local perspective, the authors wanted to draw attention to wider issues and contribute to increasing social awareness of the environmental consequences of the life cycle of the considered renewable energy sources. Additionally, it can be noticed that the world literature lacks studies in which analyzes of the life cycle of wind and photovoltaic power plants would be performed with the use of a relatively new method ReCiPe 2016. Most of the research conducted focuses only on the impact of the life cycle of the plant on GWP (Global Warming Potential), ignoring other negative impacts on the quality of the environment and human health and the depletion of raw materials, which also require detailed analyzes, especially in the perspective of sustainable development of energy systems. Therefore, the last section of the article has been renamed from "Summary and conclusions" to "Summary and discussion" and a discussion on the life cycle analyzes of wind and photovoltaic power plants has been added:

Lines 591-647: LCA analyzes in the field of wind energy initially assumed power plants with a capacity of less than one MW as their research subject. Schleisner [67] conducted one of the first studies of this type for a 500 kW turbine, while Ardente et al. [68] performed an analysis for a wind farm consisting of 11 turbines with a capacity of 660 kW each. However, several analyzes were also carried out for wind energy systems with high installed capacity, e.g. Alexandra et al. performed LCA tests for two onshore and two offshore wind power plants [69]. There are also studies devoted to local issues: Martínez et al. [70] studied the impact of the wind power plant life cycle on the environment in Spain, Wagner et al. [71] – in Germany, Schleisner [67] – in Denmark, Ardente et al. [68] – in Italy, Al-Behadili and El-Osta [72] – in Libia, Kabir et al. [73] – in Canada, Alsaleh et al. [74] – in United States, Vargas et al. [75] – in Mexico, and Oebelsa et al. [76] – in Brasil. In the case of this study, local conditions for Poland were taken into account. However, there are very little studies in the world literature in which the analyzes of the life cycle of wind power plants would be performed with the use of a relatively new method ReCiPe 2016. Most of the research conducted focuses only on the impact of the life cycle of the power plant on GWP (Global Warming Potential), ignoring other negative impacts on the quality of the environment and human health and the depletion of raw materials, which also require detailed analyzes, especially in the perspective of sustainable development of energy systems. Kabir et al. [73] examined three models of wind turbines of different power, discovering that the higher the power, the lower the CO2 emissions per kWh of produced energy. Oebels et al. [76] investigated a 1.5 MW power plant and found that the life cycle of its steel tower was mainly responsible for the highest greenhouse gas emissions. Chipindula et al. [77] performed LCA of offshore and onshore wind power plants with different installed capacity, obtaining results confirming that its increase translates into a reduction in carbon dioxide emissions per amount of electricity generated. Going further Alsaleh et al. [74] analyzed a 2 MW turbine, considering different periods of operation of this type of facilities, reaching the conclusion that the production stage is causing the highest GHG emissions to the atmosphere.

In the case of LCA analyzes conducted for photovoltaic systems, a similar trend can be observed as for wind energy systems. The subject of their research are usually various types of materials from which PV modules are produced. The largest number of analyzes was devoted to elements made of silicon, e.g. Alsema [78], Frankl et al. [79], Fthenakis and Kim [80], Dones and Frischknecht [81] also Kato et al. [82] studied the lifecycle of modules made with single-crystalline silicon (sc-Si), and on the other hand the analyzes of Alsema [78], Fthenakis and Alsema [83], Fthenakis and Kim [80], Dones and Frischknecht [81], Ito et al. [84, 85], Kato et al. [82], Nomura et al. [86] also Oliver and Jackson [87] focused on multi-crystalline silicon (mc-Si) while the research conducted by Alsema [78], Ito et al. [85], Kato et al. [82] included amorphous-silicon (a-Si), while Bravi et al. [88] assessed modules made of multi-junction thin-film silicon (µc-Si). In the literature, can also be found works devoted to other materials, e.g., research Fthenakis and Alsema [83], Fthenakis and Kim [80] also Ito et al. [85] were devoted to PV modules with cadmium telluride (CdTe), while Bravi et al. [88] analyzed the modules made from copper-indium-gallium-diselenide (CIGS), Greijer et al. [89] – dye-sensitized solar cells (DSSC). However, several studies were also carried out for high-power photovoltaic systems, e.g., Schaefer and Hagedorn [90] (2.5 MW, cells sc-Si, mc-Si and a-Si), Kato et al. [91] (10, 30 and 100 MW, cells mc-Si and a-Si) or Kato et al. [92] (10, 30 and 100 MW, cells CdTe). As in the case of wind energy power plants, also for photovoltaic power plants you can find studies devoted to local issues: Schaefer and Hagedorn [90] studied the environmental impact of the life cycle of PV systems in Germany, Dones and Frischknecht [81] – in Switzerland, Alsema [78] – in Netherlands, Bravi et al. [88] and Frankl et al. [79] – in Italy, Fthenakis and Kim [80] – in United States, Kato et al. [82, 91, 92] and Nomura et al. [86] – in Japan, Ito at el. [84, 85] – in China. In the world literature on the assessment of the life cycle of photovoltaic power plants, there are very little studies in which the analyzes would be performed using the method ReCiPe 2016. Most of the research conducted is focused on assessing the amount of CO2 and other greenhouse gas emissions [7892]. Other impacts lowering the quality of ecosystems, posing a threat to human health and exacerbating the depletion of raw materials resources, are usually not taken into account.

We apologize for the inconvenience and hope that we were able to justify the novelty of this article.

  1. The conclusion is not conclusive. The authors just repeat the method and experimental results without drawing any conclusion.

Sincerely thank you for this valuable and methodological remark.

Major changes have been made to the last section of the article.

The section summarizing the most important results has been shortened:

Lines 555-590: The main goal of the study was achieved thanks to the assessment of the life cycle of a wind and photovoltaic power plant in the context of sustainable development of energy systems.

The lifecycle of the wind power plant was distinguished by a higher potential total negative environmental impact compared to the lifecycle of the photovoltaic power plant (Table 1). Within most impact categories, the impact of the wind power plant lifecycle resulted in more harmful environmental impacts compared to the lifecycle of the photovoltaic power plant.

In the case of impact category global warming, human health, carbon dioxide had the largest share in the total emission of harmful substances (which was conditioned by significant energy inputs from conventional sources during the production of materials and elements of both analyzed technical objects) (Table 2). As in the previous impact category, carbon dioxide had the greatest impact on the level of harmful emissions in the case of global warming, terrestrial ecosystems (Table 3). As part of the impact category fine particulate matter formation, in the life cycle of both considered renewable energy sources installations, the highest level of emissions hazardous to the environment was characterized by three substances: particulates < 2.5 mm, sulfur and nitrogen oxides (Table 4). Both for the wind power plant and the photovoltaic power plant, chromium emissions to water and the atmosphere had a key influence on the development of the total level of harmful environmental effects in the area of impact category human carcinogenic toxicity. (Table 5). On the other hand, in the case of impact category human non-carcinogenic toxicity, the highest emission level in the life cycles of both technical objects was recorded for arsenic and lead (Table 6).

The highest level of potential negative environmental effects, both in the case of wind power plant and photovoltaic power plant, was recorded for areas of influence related to harmful effects on human health (Figure 6). In the lifecycle of both considered renewable energy systems, the impact on human health accounts for approx. 94% of all negative impacts (Figure 7). The substances causing the most destructive consequences in this area include carbon dioxide, particulates < 2.5 mm and sulfur dioxide (Table 7).

In the case of both analyzed life cycles, emissions to the atmosphere were the cause of the greatest number of negative environmental consequences (over 90% of all types of emissions). Substances getting into the atmospheric environment cause the most harmful effects in the area of deteriorating human health (Figure 8). In the case of emissions to the atmosphere, both in the life cycle of the wind power plant and the photovoltaic power plant, the most potential negative environmental consequences were recorded for impact categories global warming, human health and fine particulate matter formation (Table 7).

The last section has been renamed from "Summary and conclusions" to "Summary and discussion". It was supplemented with a literature review and a discussion of the obtained results, including 26 new literature items (items 67-92 in the list of references at the end of the article).

Lines 591-647: LCA analyzes in the field of wind energy initially assumed power plants with a capacity of less than one MW as their research subject. Schleisner [67] conducted one of the first studies of this type for a 500 kW turbine, while Ardente et al. [68] performed an analysis for a wind farm consisting of 11 turbines with a capacity of 660 kW each. However, several analyzes were also carried out for wind energy systems with high installed capacity, e.g. Alexandra et al. performed LCA tests for two onshore and two offshore wind power plants [69]. There are also studies devoted to local issues: Martínez et al. [70] studied the impact of the wind power plant life cycle on the environment in Spain, Wagner et al. [71] – in Germany, Schleisner [67] – in Denmark, Ardente et al. [68] – in Italy, Al-Behadili and El-Osta [72] – in Libia, Kabir et al. [73] – in Canada, Alsaleh et al. [74] – in United States, Vargas et al. [75] – in Mexico, and Oebelsa et al. [76] – in Brasil. In the case of this study, local conditions for Poland were taken into account. However, there are very little studies in the world literature in which the analyzes of the life cycle of wind power plants would be performed with the use of a relatively new method ReCiPe 2016. Most of the research conducted focuses only on the impact of the life cycle of the power plant on GWP (Global Warming Potential), ignoring other negative impacts on the quality of the environment and human health and the depletion of raw materials, which also require detailed analyzes, especially in the perspective of sustainable development of energy systems. Kabir et al. [73] examined three models of wind turbines of different power, discovering that the higher the power, the lower the CO2 emissions per kWh of produced energy. Oebels et al. [76] investigated a 1.5 MW power plant and found that the life cycle of its steel tower was mainly responsible for the highest greenhouse gas emissions. Chipindula et al. [77] performed LCA of offshore and onshore wind power plants with different installed capacity, obtaining results confirming that its increase translates into a reduction in carbon dioxide emissions per amount of electricity generated. Going further Alsaleh et al. [74] analyzed a 2 MW turbine, considering different periods of operation of this type of facilities, reaching the conclusion that the production stage is causing the highest GHG emissions to the atmosphere.

In the case of LCA analyzes conducted for photovoltaic systems, a similar trend can be observed as for wind energy systems. The subject of their research are usually various types of materials from which PV modules are produced. The largest number of analyzes was devoted to elements made of silicon, e.g. Alsema [78], Frankl et al. [79], Fthenakis and Kim [80], Dones and Frischknecht [81] also Kato et al. [82] studied the lifecycle of modules made with single-crystalline silicon (sc-Si), and on the other hand the analyzes of Alsema [78], Fthenakis and Alsema [83], Fthenakis and Kim [80], Dones and Frischknecht [81], Ito et al. [84, 85], Kato et al. [82], Nomura et al. [86] also Oliver and Jackson [87] focused on multi-crystalline silicon (mc-Si) while the research conducted by Alsema [78], Ito et al. [85], Kato et al. [82] included amorphous-silicon (a-Si), while Bravi et al. [88] assessed modules made of multi-junction thin-film silicon (µc-Si). In the literature, can also be found works devoted to other materials, e.g., research Fthenakis and Alsema [83], Fthenakis and Kim [80] also Ito et al. [85] were devoted to PV modules with cadmium telluride (CdTe), while Bravi et al. [88] analyzed the modules made from copper-indium-gallium-diselenide (CIGS), Greijer et al. [89] – dye-sensitized solar cells (DSSC). However, several studies were also carried out for high-power photovoltaic systems, e.g., Schaefer and Hagedorn [90] (2.5 MW, cells sc-Si, mc-Si and a-Si), Kato et al. [91] (10, 30 and 100 MW, cells mc-Si and a-Si) or Kato et al. [92] (10, 30 and 100 MW, cells CdTe). As in the case of wind energy power plants, also for photovoltaic power plants you can find studies devoted to local issues: Schaefer and Hagedorn [90] studied the environmental impact of the life cycle of PV systems in Germany, Dones and Frischknecht [81] – in Switzerland, Alsema [78] – in Netherlands, Bravi et al. [88] and Frankl et al. [79] – in Italy, Fthenakis and Kim [80] – in United States, Kato et al. [82, 91, 92] and Nomura et al. [86] – in Japan, Ito at el. [84, 85] – in China. In the world literature on the assessment of the life cycle of photovoltaic power plants, there are very little studies in which the analyzes would be performed using the method ReCiPe 2016. Most of the research conducted is focused on assessing the amount of CO2 and other greenhouse gas emissions [7892]. Other impacts lowering the quality of ecosystems, posing a threat to human health and exacerbating the depletion of raw materials resources, are usually not taken into account.

  1. The first three terms in points are almost the same thing.

Thank you very much for attention.

As mentioned in response to e.g., note 4, there have been major changes to the last section of this article (discussed above).

  1. The manuscript needs careful proofreading to further improve the English language.

Thank you very much for this valuable remark.

The article was revised in terms of increasing the linguistic correctness and corrections were made throughout the text.

  1. Other details should be checked again, including but not limited to the spelling, mathematical expressions and grammar.

Thank you for your attention.

The text was carefully checked for spelling, mathematical expressions, grammar, etc. Consequently, corrections were made throughout the article.

Reviewer 3 Report

The manuscript in the present form is not ready for publication. The main points are:

1.     The introduction section needs a major improvement. No LCA or similar studies on wind and PV plants are found. In addition, the novelty of study should be cleared at the end of the introduction section.

2.     Check the text to avoid typos and writing issues (e.g. Poland in line 89).

3.     “Life Cycle Assessment of plastics, materials and elements of renewable energy systems is possible thanks to the use of various models, including Life Cycle Assessment.” The sentence needs revision. Please check the whole text to have clear sentences for the potential readers.

4.     “Environmental LCA has been chosen as the method of assessing the potential impact wind and photovoltaic power plants na human health, ecosystems quality and resource depletion.” Please check the whole text grammatically.

5.     The manuscript contains many duplicate sentences. The text of a research article should be written concisely. For instance the whole opening paragraph of section 2.2 could be written in two sentences. The whole text should be more concise.

6.     “The differences in the production technology of the analyzed renewable energy sources concern mainly plastics and materials used for their production.” Plastic is a material. Please revise the whole text technically.

7.     At the end of some paragraphs (e.g. see pages 2 and 3), a group referencing is seen. Group referencing is not preferred in technical writing. Each information, fact, or any other point should be directly referenced to its appropriate reference(s).

8.     The “ReCiPe 2016 model” should be reference to an appropriate reference(s).

9.     Lines 172-181 should be referred, especially for the turbine specifications. The same should be done for the next paragraph presenting the materials used in the PV plant.

10.  Silicone in PVs doesn’t have a large share but its disposal is much complicated than others. It seems missing the share of Si for PV plants is a shortcoming of the present study. It is only mentioned “data obtained from manufacturers/producers”, which one? Which generation of PV modules? And many other questions only answered by presenting the technical specifications of the considered wind and PV plants.

11.  How much accurate are the results? Any results especially obtained from simulations should be validated. In addition, no comparison with other studies is seen.

12.  The conclusion section is too lengthy. Meanwhile, no new information or interpretation should be presented in this section. Hence, it is not typical to see new references in this section.

Author Response

RESPONSE TO REVIEWER 3 COMMENTS

Dear Reviewer,

Thank you for taking the time to read our article and for the review. We appreciate all comments and suggestions as they will help us improve our research workflow in the future.

 Detailed answers to all comments are provided below. Amendments in the text of the article are marked in red.

  1. The introduction section needs a major improvement. No LCA or similar studies on wind and PV plants are found. In addition, the novelty of study should be cleared at the end of the introduction section.

Thank you for the note that puts the article in order.

The introductory section was supplemented with a justification for the novelty of the study:

Lines 89-98: There are not many studies in the world literature in which analyzes of the life cycle of wind and solar power plants would be performed using a relatively new method ReCi-Pe 2016. Most of the research conducted focuses only on the impact of the life cycle on GWP (Global Warming Potential), ignoring other negative impacts of the systems under consideration, which reduce the quality of the environment, pose a threat to human health and increase the depletion of raw materials, which also require detailed analyzes, especially in view of the sustainable development of energy systems (see section 5 for details). In Poland, unfortunately, analyzes using the LCA methodology are still not very popular. The study tries to outline the local perspective of the environmental impact of selected renewable energy sources, hence it was decided to study two real cases.

The last section has been renamed from "Summary and conclusions" to "Summary and discussion". It was supplemented with a literature review and a discussion of the obtained results, including 26 new literature items (items 67-92 in the list of references at the end of the article), in the field of life cycle analysis of wind energy and photovoltaic systems:

Lines 591-647: LCA analyzes in the field of wind energy initially assumed power plants with a capacity of less than one MW as their research subject. Schleisner [67] conducted one of the first studies of this type for a 500 kW turbine, while Ardente et al. [68] performed an analysis for a wind farm consisting of 11 turbines with a capacity of 660 kW each. However, several analyzes were also carried out for wind energy systems with high installed capacity, e.g. Alexandra et al. performed LCA tests for two onshore and two offshore wind power plants [69]. There are also studies devoted to local issues: Martínez et al. [70] studied the impact of the wind power plant life cycle on the environment in Spain, Wagner et al. [71] – in Germany, Schleisner [67] – in Denmark, Ardente et al. [68] – in Italy, Al-Behadili and El-Osta [72] – in Libia, Kabir et al. [73] – in Canada, Alsaleh et al. [74] – in United States, Vargas et al. [75] – in Mexico, and Oebelsa et al. [76] – in Brasil. In the case of this study, local conditions for Poland were taken into account. However, there are very little studies in the world literature in which the analyzes of the life cycle of wind power plants would be performed with the use of a relatively new method ReCiPe 2016. Most of the research conducted focuses only on the impact of the life cycle of the power plant on GWP (Global Warming Potential), ignoring other negative impacts on the quality of the environment and human health and the depletion of raw materials, which also require detailed analyzes, especially in the perspective of sustainable development of energy systems. Kabir et al. [73] examined three models of wind turbines of different power, discovering that the higher the power, the lower the CO2 emissions per kWh of produced energy. Oebels et al. [76] investigated a 1.5 MW power plant and found that the life cycle of its steel tower was mainly responsible for the highest greenhouse gas emissions. Chipindula et al. [77] performed LCA of offshore and onshore wind power plants with different installed capacity, obtaining results confirming that its increase translates into a reduction in carbon dioxide emissions per amount of electricity generated. Going further Alsaleh et al. [74] analyzed a 2 MW turbine, considering different periods of operation of this type of facilities, reaching the conclusion that the production stage is causing the highest GHG emissions to the atmosphere.

In the case of LCA analyzes conducted for photovoltaic systems, a similar trend can be observed as for wind energy systems. The subject of their research are usually various types of materials from which PV modules are produced. The largest number of analyzes was devoted to elements made of silicon, e.g. Alsema [78], Frankl et al. [79], Fthenakis and Kim [80], Dones and Frischknecht [81] also Kato et al. [82] studied the lifecycle of modules made with single-crystalline silicon (sc-Si), and on the other hand the analyzes of Alsema [78], Fthenakis and Alsema [83], Fthenakis and Kim [80], Dones and Frischknecht [81], Ito et al. [84, 85], Kato et al. [82], Nomura et al. [86] also Oliver and Jackson [87] focused on multi-crystalline silicon (mc-Si) while the research conducted by Alsema [78], Ito et al. [85], Kato et al. [82] included amorphous-silicon (a-Si), while Bravi et al. [88] assessed modules made of multi-junction thin-film silicon (µc-Si). In the literature, can also be found works devoted to other materials, e.g., research Fthenakis and Alsema [83], Fthenakis and Kim [80] also Ito et al. [85] were devoted to PV modules with cadmium telluride (CdTe), while Bravi et al. [88] analyzed the modules made from copper-indium-gallium-diselenide (CIGS), Greijer et al. [89] – dye-sensitized solar cells (DSSC). However, several studies were also carried out for high-power photovoltaic systems, e.g., Schaefer and Hagedorn [90] (2.5 MW, cells sc-Si, mc-Si and a-Si), Kato et al. [91] (10, 30 and 100 MW, cells mc-Si and a-Si) or Kato et al. [92] (10, 30 and 100 MW, cells CdTe). As in the case of wind energy power plants, also for photovoltaic power plants you can find studies devoted to local issues: Schaefer and Hagedorn [90] studied the environmental impact of the life cycle of PV systems in Germany, Dones and Frischknecht [81] – in Switzerland, Alsema [78] – in Netherlands, Bravi et al. [88] and Frankl et al. [79] – in Italy, Fthenakis and Kim [80] – in United States, Kato et al. [82, 91, 92] and Nomura et al. [86] – in Japan, Ito at el. [84, 85] – in China. In the world literature on the assessment of the life cycle of photovoltaic power plants, there are very little studies in which the analyzes would be performed using the method ReCiPe 2016. Most of the research conducted is focused on assessing the amount of CO2 and other greenhouse gas emissions [78–92]. Other impacts lowering the quality of ecosystems, posing a threat to human health and exacerbating the depletion of raw materials resources, are usually not taken into account.

  1. Check the text to avoid typos and writing issues (e.g. Poland in line 89).

Thank you for your attention.

Duplicate of word “Poland” is removed from line no. 89 (line 108 in new version of article).

A detailed review has been made and any typos noticed in the text have been corrected.

  1. “Life Cycle Assessment of plastics, materials and elements of renewable energy systems is possible thanks to the use of various models, including Life Cycle Assessment.” The sentence needs revision. Please check the whole text to have clear sentences for the potential readers.

Thank you very much for this attention.

The above-mentioned sentence has been revised:

Lines 108-110: Life cycle assessment of materials and elements of renewable energy systems is possible thanks to the use of various models, including Environmental LCA.

A detailed review has been made and all the potentially incomprehensible sentences that have been noticed in the text have been corrected.

  1. “Environmental LCA has been chosen as the method of assessing the potential impact wind and photovoltaic power plants na human health, ecosystems quality and resource depletion.” Please check the whole text grammatically.

Thank you very much for this valuable remark.

Changed wording from “na” on “on”:

Lines 110-112: This method has been chosen as the model of assessing the potential impact wind and photovoltaic power plants on human health, ecosystems quality and resource depletion.

A detailed review has been made and efforts have been made to correct any grammatical errors noticed in the text.

  1. The manuscript contains many duplicate sentences. The text of a research article should be written concisely. For instance the whole opening paragraph of section 2.2 could be written in two sentences. The whole text should be more concise.

Thank you for your attention.

A detailed review has been carried out and efforts have been made to remove any duplicate sentences noticed in the text.

It was decided to describe the methodology used in the research in more detail in order to bring its most important assumptions closer to the readers who encounter LCA analyzes for the first time.

Changes have been made throughout the text and regarding section 2.2 - it has been tried to be kept as short as possible, but at the same time to include the most important information on the scope of determination of goal and scope:

Lines: 135-157: The aim of the analysis that will be carried out in this study will be to compare the environmental impacts associated with the life cycle of wind and photovoltaic power plants (comparative analysis). The LCA analysis will be used to determine whether there are differences in the magnitude of the environmental impact generated during the life cycles of selected renewable energy sources operating on the basis of two different technologies [20,2830].

The systems of the analyzed technical objects were constructed in a comparable manner in terms of the depth and width of the analysis. The geographical scope is an area of Europe, as the companies that provided the data have a very strong position in the entire European market. The time range also cover the same range due both of the wind power plant and the photovoltaic power plant have a lifecycle of approximately 20 years. The cut-off level adopted for the research was 0.1%.

The conducted analysis can be classified as bottom-up and was mainly used to describe the existing reality (retrospective analysis), but also to model more pro-environmental solutions (prospective analysis). The level of advancement of the analysis classifies it among detailed analyzes. The data used in the analysis was obtained from producers of the considered renewable energy systems or from SimaPro databases.

The functional unit is the value of the installed capacity in each of the research objects – 2 MW. The environmental aspects of the assessment include twenty-two impact categories specific to the ReCiPe 2016 model (listed in Table 1). The obtained research results were additionally grouped and compiled into three areas of influence: human health, ecosystems and resources. Four areas of emission of individual chemical compounds were also specified, including: air, water, soil and raw [20,31–32].

  1. “The differences in the production technology of the analyzed renewable energy sources concern mainly plastics and materials used for their production.” Plastic is a material. Please revise the whole text technically.

Thank you very much for attention.

The word "plastic" has been removed from the sentences below:

Lines 32-35: On the basis of the obtained results, guidelines were proposed for pro-ecological changes in the life cycle of materials and elements of the considered technical facilities of renewable energy sources, aimed at better implementation of the main assumptions of contemporary sustainable development (especially in the field of environmental protection).

Lines 110-112: Life cycle assessment of materials and elements of renewable energy systems is possible thanks to the use of various models, including Environmental LCA.

Lines 183-184: The total mass of materials and elements of the tested photovoltaic power plant is around three hundred tons.

Lines 279-281: The adopted assumptions, methods used, the depth of the analysis, its detail and precision of data for both materials and wind and photovoltaic power plants elements were consistent with the goal and scope of the research.

Lines 328-331: In both cases, the largest share in the total emission was carbon dioxide (W – 2.73×104 Pt and PV – 8.05×103 Pt), which is caused by a significantly high energy input from conventional sources during the production of materials and components of both analyzed technical objects.

Lines 444-446: This is the cause of significant emissions of this element in the lifecycle of winds and photovoltaic power plants (mainly as part of the production processes of materials and elements of these technical facilities).

Lines 477-481 It is part of the carbon cycle in nature, it is a product of combustion and respiration (hence its significant amount is emitted mainly in the processes of producing materials and elements of wind and photovoltaic power plants, which are characterized by significant energy and material consumption).

Lines 552-554: It is visible, therefore, that the analyzed technical objects of renewable energy are made of materials and elements, largely characterized by high energy and material consumption in production processes.

Lines 563-566: In the case of impact category global warming, human health, carbon dioxide had the largest share in the total emission of harmful substances (which was conditioned by significant energy inputs from conventional sources during the production of materials and elements of both analyzed technical objects).

Lines 657-658: It is also important to maximize the level of recovery and re-use of other substances and materials used in all technological processes.

Lines 660-662: Currently, it is also obvious that we strive to increase the material and energy efficiency of the processes of producing materials and components, and to reduce the amount and range of emissions of the generated pollutants.

A detailed review has been carried out and efforts have been made to correct any errors of a technical nature noted in the text.

  1. At the end of some paragraphs (e.g. see pages 2 and 3), a group referencing is seen. Group referencing is not preferred in technical writing. Each information, fact, or any other point should be directly referenced to its appropriate reference(s).

Sincerely thank you for this remark.

Some scientific articles refer to a group. It was decided to use this form because in all literature items included in a given group you can find information (which was used) related to the topic presented within a specific paragraph of the text.

  1. The “ReCiPe 2016 model” should be reference to an appropriate reference(s).

Thank you for your attention.

As a result of a mistake, three literature sources were incorrectly assigned to the "ReCiPe 2016 model". The following changes were made to the reference list :

  1. BaÅ‚dowska-Witos, P.; Kruszelnicka, W.; Kasner, R.; Rudnicki, J.; Tomporowski, A; Flizikowski, J. Impact of the plastic bottle production on the natural environment. Part 1. Application of the ReCiPe 2016 assessment method to identify environmental problems Przem. Chem. 2019, 10, 1662–1667. [doi: 10.15199/62.2019.10.27]
  2. Dekker, E.; Zijp, M.C.; van de Kamp, M.E.; Temme, E.H.M.; van Zelm, R. A taste of the new ReCiPe for life cycle assessment: consequences of the updated impact assessment method on food product LCAs Int. J. Life Cycle Assess 2020, 25, 2315–2324. [doi: 10.1007/s11367-019-01653-3]
  3. Rashedi, A.; Khanam, T. Life cycle assessment of most widely adopted solar photovoltaic energy technologies by mid-point and end-point indicators of ReCiPe method Environ Sci Pollut Res 2020, 27, 29075–29090. [doi: 10.1007/s11356-020-09194-1]

  1. Lines 172-181 should be referred, especially for the turbine specifications. The same should be done for the next paragraph presenting the materials used in the PV plant.

Thank you very much for this attention.

In the indicated fragments, a reference to the data source regarding the specification of a wind farm and a photovoltaic plant has been added:

Lines 181-182: (…) (data obtained from the investor and producers).

Lines 190-191: (…) (data obtained from the investor and producers).

  1. Silicone in PVs doesn’t have a large share but its disposal is much complicated than others. It seems missing the share of Si for PV plants is a shortcoming of the present study. It is only mentioned “data obtained from manufacturers/producers”, which one? Which generation of PV modules? And many other questions only answered by presenting the technical specifications of the considered wind and PV plants.

Thank you very much for this valuable remark.

The type of photovoltaic modules has been added to the characteristics of the analyzed photovoltaic installation (single-crystalline silicon photovoltaic modules):

Lines 184-186: Single-crystalline silicon photovoltaic modules have the largest share in the mass of the object – approx. 62% (including approx. 47% of their mass is solar glass, and approx. 45% - aluminum).

Unfortunately, due to the concluded confidentiality agreements, the detailed technical specification of a wind and photovoltaic power plant could not be presented in this article. This is mentioned in the passage:

Lines 168-171: Due to the conclusion of a data confidentiality agreement with companies producing the analyzed renewable energy systems, all detailed information on the structure of the analysis objects and technological data are not disclosed in this study.

  1. How much accurate are the results? Any results especially obtained from simulations should be validated. In addition, no comparison with other studies is seen.

Thank you for your attention.

The issue of data quality (in general) is discussed in section 2.3:

Lines 165-168: Information on key processes was obtained directly from manufacturers of materials and components. Data on less significant processes and materials from the point of view of environmental impact were obtained from databases included in the SimaPro 9.3 software (Ecoinvent 3.8 database).

In accordance with ISO standards, the obtained data was accurate and precise, was complete and obtained from sources deemed reliable by the authors.

For the analyzed objects of analysis, it is not possible to validate the results in the form of a comparative Monte Carlo analysis, due to too large discrepancies in the technologies used at each stage of their life cycle.

As mentioned before, the last section has been renamed from "Summary and conclusions" to "Summary and discussion". It was supplemented with a literature review and a discussion of the obtained results, including 26 new literature items (items 67-92 in the list of references at the end of the article), in the field of life cycle analysis of wind energy and photovoltaic systems:

Lines 591-647: LCA analyzes in the field of wind energy initially assumed power plants with a capacity of less than one MW as their research subject. Schleisner [67] conducted one of the first studies of this type for a 500 kW turbine, while Ardente et al. [68] performed an analysis for a wind farm consisting of 11 turbines with a capacity of 660 kW each. However, several analyzes were also carried out for wind energy systems with high installed capacity, e.g. Alexandra et al. performed LCA tests for two onshore and two offshore wind power plants [69]. There are also studies devoted to local issues: Martínez et al. [70] studied the impact of the wind power plant life cycle on the environment in Spain, Wagner et al. [71] – in Germany, Schleisner [67] – in Denmark, Ardente et al. [68] – in Italy, Al-Behadili and El-Osta [72] – in Libia, Kabir et al. [73] – in Canada, Alsaleh et al. [74] – in United States, Vargas et al. [75] – in Mexico, and Oebelsa et al. [76] – in Brasil. In the case of this study, local conditions for Poland were taken into account. However, there are very little studies in the world literature in which the analyzes of the life cycle of wind power plants would be performed with the use of a relatively new method ReCiPe 2016. Most of the research conducted focuses only on the impact of the life cycle of the power plant on GWP (Global Warming Potential), ignoring other negative impacts on the quality of the environment and human health and the depletion of raw materials, which also require detailed analyzes, especially in the perspective of sustainable development of energy systems. Kabir et al. [73] examined three models of wind turbines of different power, discovering that the higher the power, the lower the CO2 emissions per kWh of produced energy. Oebels et al. [76] investigated a 1.5 MW power plant and found that the life cycle of its steel tower was mainly responsible for the highest greenhouse gas emissions. Chipindula et al. [77] performed LCA of offshore and onshore wind power plants with different installed capacity, obtaining results confirming that its increase translates into a reduction in carbon dioxide emissions per amount of electricity generated. Going further Alsaleh et al. [74] analyzed a 2 MW turbine, considering different periods of operation of this type of facilities, reaching the conclusion that the production stage is causing the highest GHG emissions to the atmosphere.

In the case of LCA analyzes conducted for photovoltaic systems, a similar trend can be observed as for wind energy systems. The subject of their research are usually various types of materials from which PV modules are produced. The largest number of analyzes was devoted to elements made of silicon, e.g. Alsema [78], Frankl et al. [79], Fthenakis and Kim [80], Dones and Frischknecht [81] also Kato et al. [82] studied the lifecycle of modules made with single-crystalline silicon (sc-Si), and on the other hand the analyzes of Alsema [78], Fthenakis and Alsema [83], Fthenakis and Kim [80], Dones and Frischknecht [81], Ito et al. [84, 85], Kato et al. [82], Nomura et al. [86] also Oliver and Jackson [87] focused on multi-crystalline silicon (mc-Si) while the research conducted by Alsema [78], Ito et al. [85], Kato et al. [82] included amorphous-silicon (a-Si), while Bravi et al. [88] assessed modules made of multi-junction thin-film silicon (µc-Si). In the literature, can also be found works devoted to other materials, e.g., research Fthenakis and Alsema [83], Fthenakis and Kim [80] also Ito et al. [85] were devoted to PV modules with cadmium telluride (CdTe), while Bravi et al. [88] analyzed the modules made from copper-indium-gallium-diselenide (CIGS), Greijer et al. [89] – dye-sensitized solar cells (DSSC). However, several studies were also carried out for high-power photovoltaic systems, e.g., Schaefer and Hagedorn [90] (2.5 MW, cells sc-Si, mc-Si and a-Si), Kato et al. [91] (10, 30 and 100 MW, cells mc-Si and a-Si) or Kato et al. [92] (10, 30 and 100 MW, cells CdTe). As in the case of wind energy power plants, also for photovoltaic power plants you can find studies devoted to local issues: Schaefer and Hagedorn [90] studied the environmental impact of the life cycle of PV systems in Germany, Dones and Frischknecht [81] – in Switzerland, Alsema [78] – in Netherlands, Bravi et al. [88] and Frankl et al. [79] – in Italy, Fthenakis and Kim [80] – in United States, Kato et al. [82, 91, 92] and Nomura et al. [86] – in Japan, Ito at el. [84, 85] – in China. In the world literature on the assessment of the life cycle of photovoltaic power plants, there are very little studies in which the analyzes would be performed using the method ReCiPe 2016. Most of the research conducted is focused on assessing the amount of CO2 and other greenhouse gas emissions [7892]. Other impacts lowering the quality of ecosystems, posing a threat to human health and exacerbating the depletion of raw materials resources, are usually not taken into account.

  1. The conclusion section is too lengthy. Meanwhile, no new information or interpretation should be presented in this section. Hence, it is not typical to see new references in this section.

Thank you very much for attention.

Renamed the last section of the article from "Summary and conclusions" to "Summary and discussion”.

The section summarizing the most important results has been shortened:

Lines 555-590: The main goal of the study was achieved thanks to the assessment of the life cycle of a wind and photovoltaic power plant in the context of sustainable development of energy systems.

The lifecycle of the wind power plant was distinguished by a higher potential total negative environmental impact compared to the lifecycle of the photovoltaic power plant (Table 1). Within most impact categories, the impact of the wind power plant lifecycle resulted in more harmful environmental impacts compared to the lifecycle of the photovoltaic power plant.

In the case of impact category global warming, human health, carbon dioxide had the largest share in the total emission of harmful substances (which was conditioned by significant energy inputs from conventional sources during the production of materials and elements of both analyzed technical objects) (Table 2). As in the previous impact category, carbon dioxide had the greatest impact on the level of harmful emissions in the case of global warming, terrestrial ecosystems (Table 3). As part of the impact category fine particulate matter formation, in the life cycle of both considered renewable energy sources installations, the highest level of emissions hazardous to the environment was characterized by three substances: particulates < 2.5 mm, sulfur and nitrogen oxides (Table 4). Both for the wind power plant and the photovoltaic power plant, chromium emissions to water and the atmosphere had a key influence on the development of the total level of harmful environmental effects in the area of impact category human carcinogenic toxicity. (Table 5). On the other hand, in the case of impact category human non-carcinogenic toxicity, the highest emission level in the life cycles of both technical objects was recorded for arsenic and lead (Table 6).

The highest level of potential negative environmental effects, both in the case of wind power plant and photovoltaic power plant, was recorded for areas of influence related to harmful effects on human health (Figure 6). In the lifecycle of both considered renewable energy systems, the impact on human health accounts for approx. 94% of all negative impacts (Figure 7). The substances causing the most destructive consequences in this area include carbon dioxide, particulates < 2.5 mm and sulfur dioxide (Table 7).

In the case of both analyzed life cycles, emissions to the atmosphere were the cause of the greatest number of negative environmental consequences (over 90% of all types of emissions). Substances getting into the atmospheric environment cause the most harmful effects in the area of deteriorating human health (Figure 8). In the case of emissions to the atmosphere, both in the life cycle of the wind power plant and the photovoltaic power plant, the most potential negative environmental consequences were recorded for impact categories global warming, human health and fine particulate matter formation (Table 7).

As mentioned before, the section was supplemented by with a literature review and a discussion of the obtained results, including 26 new literature items (items 67-92 in the list of references at the end of the article), in the field of life cycle analysis of wind energy and photovoltaic systems:

Lines 591-647: LCA analyzes in the field of wind energy initially assumed power plants with a capacity of less than one MW as their research subject. Schleisner [67] conducted one of the first studies of this type for a 500 kW turbine, while Ardente et al. [68] performed an analysis for a wind farm consisting of 11 turbines with a capacity of 660 kW each. However, several analyzes were also carried out for wind energy systems with high installed capacity, e.g. Alexandra et al. performed LCA tests for two onshore and two offshore wind power plants [69]. There are also studies devoted to local issues: Martínez et al. [70] studied the impact of the wind power plant life cycle on the environment in Spain, Wagner et al. [71] – in Germany, Schleisner [67] – in Denmark, Ardente et al. [68] – in Italy, Al-Behadili and El-Osta [72] – in Libia, Kabir et al. [73] – in Canada, Alsaleh et al. [74] – in United States, Vargas et al. [75] – in Mexico, and Oebelsa et al. [76] – in Brasil. In the case of this study, local conditions for Poland were taken into account. However, there are very little studies in the world literature in which the analyzes of the life cycle of wind power plants would be performed with the use of a relatively new method ReCiPe 2016. Most of the research conducted focuses only on the impact of the life cycle of the power plant on GWP (Global Warming Potential), ignoring other negative impacts on the quality of the environment and human health and the depletion of raw materials, which also require detailed analyzes, especially in the perspective of sustainable development of energy systems. Kabir et al. [73] examined three models of wind turbines of different power, discovering that the higher the power, the lower the CO2 emissions per kWh of produced energy. Oebels et al. [76] investigated a 1.5 MW power plant and found that the life cycle of its steel tower was mainly responsible for the highest greenhouse gas emissions. Chipindula et al. [77] performed LCA of offshore and onshore wind power plants with different installed capacity, obtaining results confirming that its increase translates into a reduction in carbon dioxide emissions per amount of electricity generated. Going further Alsaleh et al. [74] analyzed a 2 MW turbine, considering different periods of operation of this type of facilities, reaching the conclusion that the production stage is causing the highest GHG emissions to the atmosphere.

In the case of LCA analyzes conducted for photovoltaic systems, a similar trend can be observed as for wind energy systems. The subject of their research are usually various types of materials from which PV modules are produced. The largest number of analyzes was devoted to elements made of silicon, e.g. Alsema [78], Frankl et al. [79], Fthenakis and Kim [80], Dones and Frischknecht [81] also Kato et al. [82] studied the lifecycle of modules made with single-crystalline silicon (sc-Si), and on the other hand the analyzes of Alsema [78], Fthenakis and Alsema [83], Fthenakis and Kim [80], Dones and Frischknecht [81], Ito et al. [84, 85], Kato et al. [82], Nomura et al. [86] also Oliver and Jackson [87] focused on multi-crystalline silicon (mc-Si) while the research conducted by Alsema [78], Ito et al. [85], Kato et al. [82] included amorphous-silicon (a-Si), while Bravi et al. [88] assessed modules made of multi-junction thin-film silicon (µc-Si). In the literature, can also be found works devoted to other materials, e.g., research Fthenakis and Alsema [83], Fthenakis and Kim [80] also Ito et al. [85] were devoted to PV modules with cadmium telluride (CdTe), while Bravi et al. [88] analyzed the modules made from copper-indium-gallium-diselenide (CIGS), Greijer et al. [89] – dye-sensitized solar cells (DSSC). However, several studies were also carried out for high-power photovoltaic systems, e.g., Schaefer and Hagedorn [90] (2.5 MW, cells sc-Si, mc-Si and a-Si), Kato et al. [91] (10, 30 and 100 MW, cells mc-Si and a-Si) or Kato et al. [92] (10, 30 and 100 MW, cells CdTe). As in the case of wind energy power plants, also for photovoltaic power plants you can find studies devoted to local issues: Schaefer and Hagedorn [90] studied the environmental impact of the life cycle of PV systems in Germany, Dones and Frischknecht [81] – in Switzerland, Alsema [78] – in Netherlands, Bravi et al. [88] and Frankl et al. [79] – in Italy, Fthenakis and Kim [80] – in United States, Kato et al. [82, 91, 92] and Nomura et al. [86] – in Japan, Ito at el. [84, 85] – in China. In the world literature on the assessment of the life cycle of photovoltaic power plants, there are very little studies in which the analyzes would be performed using the method ReCiPe 2016. Most of the research conducted is focused on assessing the amount of CO2 and other greenhouse gas emissions [7892]. Other impacts lowering the quality of ecosystems, posing a threat to human health and exacerbating the depletion of raw materials resources, are usually not taken into account.

The authors intended the new references in the last section to complement the obtained results of the analyzes. This is not a typical solution, but it is found in some scientific studies.

Reviewer 4 Report

According to my opinion, the manuscript needs to be restructured. It is very difficult to follow due to many repetitions, e.g., "ReCiPe 2016", "SimaPro", or ”22” are mentioned many times, and then, the section “Summary and conclusions” where all is repeated again and again. Just explain method and used software at the begging of Materials and Methods and that is enough. Besides, in the section Results, many paragraphs begin with some obtained numbers/results and continue with some general facts about negative impacts on the health/environment in general and without any appropriate literature citations (e.g., lines 319-336, 348-359, 372-379, etc.,).  The section “Summary and conclusions” should be removed. A section “Discussion” should be created where the results will be considered in the light of comparison with the other studies, not only with similar methodology, but in the relation to sustainable development of energy systems at all, …. , discuss the strengths and shortcomings of used method and calculation procedure, etc. If you just get some data and put it in is some software and then show outputs there is no point. Give “deeper” meaning to the results and how much of a negative impact it really is (numbers by themselves mean nothing).

The meaning of the unit “Pt” was given poorly. Use term “ecopoint” instead “environmental point”.

Major revision is recommended.

Author Response

RESPONSE TO REVIEWER 4 COMMENTS

Dear Reviewer,

Thank you for taking the time to read our article and for the review. We appreciate all comments and suggestions as they will help us improve our research workflow in the future.

 Detailed answers to all comments are provided below. Amendments in the text of the article are marked in red.

  1. According to my opinion, the manuscript needs to be restructured. It is very difficult to follow due to many repetitions, e.g., "ReCiPe 2016", "SimaPro", or ”22” are mentioned many times, and then, the section “Summary and conclusions” where all is repeated again and again. Just explain method and used software at the begging of Materials and Methods and that is enough. Besides, in the section Results, many paragraphs begin with some obtained numbers/results and continue with some general facts about negative impacts on the health/environment in general and without any appropriate literature citations (e.g., lines 319-336, 348-359, 372-379, etc.,).

Thank you very much for this valuable remark.

The additional methodological clarifications in the last section of the article "Summary and discussion”.

Additionally, efforts were made to shorten the methodology description in section 2 „Materials and Methods”. For example, section 2.2 has been shortened in such a way as to simultaneously include the most important information in the field of determination of goal and scope:

Lines: 135-157: The aim of the analysis that will be carried out in this study will be to compare the environmental impacts associated with the life cycle of wind and photovoltaic power plants (comparative analysis). The LCA analysis will be used to determine whether there are differences in the magnitude of the environmental impact generated during the life cycles of selected renewable energy sources operating on the basis of two different technologies [20,2830].

The systems of the analyzed technical objects were constructed in a comparable manner in terms of the depth and width of the analysis. The geographical scope is an area of Europe, as the companies that provided the data have a very strong position in the entire European market. The time range also cover the same range due both of the wind power plant and the photovoltaic power plant have a lifecycle of approximately 20 years. The cut-off level adopted for the research was 0.1%.

The conducted analysis can be classified as bottom-up and was mainly used to describe the existing reality (retrospective analysis), but also to model more pro-environmental solutions (prospective analysis). The level of advancement of the analysis classifies it among detailed analyzes. The data used in the analysis was obtained from producers of the considered renewable energy systems or from SimaPro databases.

The functional unit is the value of the installed capacity in each of the research objects – 2 MW. The environmental aspects of the assessment include twenty-two impact categories specific to the ReCiPe 2016 model (listed in Table 1). The obtained research results were additionally grouped and compiled into three areas of influence: human health, ecosystems and resources. Four areas of emission of individual chemical compounds were also specified, including: air, water, soil and raw [20,31–32].

The same was done with the rest of the text in section 2.

Changes were made to section 3 "Results" - it was supplemented with additional references to the literature and issues that were considered likely to raise doubts by readers were clarified, e.g.:

In section 3 the description of the PV module production process was extended, with the greatest energy and the production of silicon monocrystals by the Czochralski method is characterized by material absorption:

Lines 305-321: In most of the analyzed impact categories, the impact of the wind power plant life cycle resulted in more potential negative environmental consequences compared to the life cycle of the photovoltaic power plant. Two were the exceptions impact categories: ionizing radiation (PV – 1.12×10-1 Pt and W – 9.13×10-2 Pt) and marine ecotoxicity (PV – 1.69×100 Pt and W – 6.27×10-1 Pt), which is most likely due to the specificity and high energy consumption of the photovoltaic modules production processes. In the production of photovoltaic modules, the most long-lasting and energy-consuming process is the cultivation of monocrystalline silicon crystals. The first step is the production of pure silicon from silicon dioxide by chemical methods. The material is then melted and then crystallized by cooling. The single crystal must not contain foreign atoms, so the process must take place under special conditions. For this purpose, a vacuum furnace is usually used, in which crystallization takes place without the access of gases, in particular oxygen (this method was developed by J. Czochralski). After removing from the furnace, the crystals are mechanically processed in order to obtain the highest quality silicon wafers, necessary for the production of photovoltaic modules. The life cycle of a wind power plant is characterized by a higher potential total negative impact on the environment (in total 8.98×104 Pt) than the life cycle of a photovoltaic power plant (total 3.13×104 Pt) (Table 1) [59].

References no. 59: Aleksic, J.; Zielke, P.; Szymczyk, J.A. Temperature and Flow Visualization in a Simulation of the Czochralski Process Using Temperature-Sensitive Liquid Crystals. Ann. N. Y. Acad. Sci. 2002, 972, 158–163.

Information on the impact of methane on the environment:

Lines 325-348: As part of the impact category global warming, human health, the wind power plant life cycle resulted in a greater emission of chemical compounds hazardous to health (total impact at the level of 3.06×104 Pt) compared to the lifecycle of the photovoltaic power plant (total 9.53×103 Pt). In both cases, the largest share in the total emission was carbon dioxide (W – 2.73×104 Pt and PV – 8.05×103 Pt), which is caused by a significantly high energy input from conventional sources during the production of materials and components of both analyzed technical objects. The remaining greenhouse gases causing the most negative consequences in the life cycle of the wind power plant include methane (2.98×103 Pt) and dinitrogen monoxide (1.98×102 Pt), and in the lifecycle of the photovoltaic power plant - also methane (1.09×103 Pt) and tetrafluoromethane , CFC-14 (1.87×102 Pt) (Table 2). Methane, which is also classified as GHG, is mentioned much less frequently than in the case of carbon dioxide. However, it is a chemical compound that is dangerous to the environment. During the first 20 years in the atmosphere, the climatic effect of one ton of methane is about 85 times greater than one ton of carbon dioxide. The two largest sources of methane are the energy industry and agriculture. Methane from the energy industry is primarily emitted during the extraction, transport and storage of fossil fuels such as coal, oil and natural gas. Poland is the largest emitter of fossil methane in Europe. Methane, above all, strongly pollutes the air that people breathe. This can consequently lead to diseases such as asthma and emphysema. Current reports of the Intergovernmental Panel on Climate Change (IPCC) indicate that the global methane emissions should be reduced by approx. 50% in the next 20 years. Specific reduction steps have not been indicated, but the significant role of the energy sector in this regard is underlined. The key solution seems to be to abandon fossil fuels as quickly as possible and ensure that methane does not leak from closed mines or from abandoned gas or oil wells [60].

References no. 60: Howarth, R.W.; Jacobson, M.Z. How green is blue hydrogen? Energy Sci Eng. 2021, 9, 1676–1687.

Information on the impact of nitrogen oxides on the environment and human health:

Lines 352-371: Also in the case of impact category global warming, terrestrial ecosystems, the wind power plant life cycle causes more negative environmental consequences in the considered scope (total impact equal to 3.06×103 Pt) compared to the lifecycle of the photovoltaic power plant (total 9.55×102 Pt). Similarly, to the previously discussed impact category, carbon dioxide (W – 2.73×103 Pt and PV – 8.08×102 Pt) has the greatest influence on the shaping of the value of harmful emissions. As part of the life cycle of the wind power plant, significant emission values were also noted: methane (2.98×102 Pt) and dinitrogen monoxide (1.98×101 Pt), while in the lifecycle of the photovoltaic power plant - tetrafluoromethane, CFC-14 (1.87×101 Pt) and dinitrogen monoxide (9.38×100 Pt) (Table 3). Dinitrogen monoxide is one of the main greenhouse gases. Nitrogen oxides are one of the most dangerous components of smog. Their toxicity is many times greater than that of carbon monoxide or sulfur dioxide. Dinitrogen monoxide, being the third most important long-term GHG, contributes significantly to global warming and is a substance that depletes stratospheric ozone significantly. Its greenhouse effect potential is approx. 140 times stronger than that of carbon dioxide. The average lifetime of this gas in the atmosphere is estimated to be over 100 years. Nitrogen oxides have a negative effect on human health. First of all, they irritate the respiratory system, posing a serious threat especially to people suffering from asthma and lung diseases - contributing to the exacerbation of ailments. Their negative impact on health also occurs indirectly, because nitrogen oxides contribute to soil acidification and the formation of carcinogenic compounds that penetrate into plants [61].

References no. 61: Manisalidis, I.; Stavropoulou, E.; Stavropoulos, A.; Bezirtzoglou, E. Environmental and Health Impacts of Air Pollution: A Review. Front. Public Health, 2020, 8, 1–13.

Information on the impact of particulate matter on human health:

Lines 375-391: Another impact category with a key impact on the development of the level of harmful effects on the environment of the life cycle of both analyzed technical objects is fine particulate matter formation. Within its framework, more negative environmental consequences were noted for the life cycle of the wind power plant (jointly 4.45×104 Pt) rather than photovoltaic power plant (jointly 1.59×104 Pt). In the cycle of both renewable energy installations mentioned above, three substances stood out with the highest level of destructive emissions: particulates < 2.5 mm (W – 2.58×104 Pt and PV – 6.43×103 Pt), sulfur dioxide (W – 1.35×104 Pt and PV – 7.15×103 Pt) and nitrogen oxides (W – 4.84×103 Pt and PV – 2.14×103 Pt) (Table 4). Particulates < 2.5 mm are atmospheric aerosols not larger than 2.5 micrometers in diameter. Particles of this size are considered to be particularly dangerous to human health because they bypass many of the body's defenses (such as nose hair and mucus) that act to trap particles before they penetrate deeper into the body. PM 2.5 particles can travel to the lungs, further into the alveoli and eventually into the bloodstream. These particles may contain toxic chemicals. This type of particulate matter is responsible, inter alia, in for worsening of asthma, decreased lung function, cancer (of the lungs, throat and larynx), abnormal heart rhythm or inflammation of the blood vessels. The main sources of particulates <2.5 mm are fossil fuel combustion, transport and industry [61].

References no. 61: Manisalidis, I.; Stavropoulou, E.; Stavropoulos, A.; Bezirtzoglou, E. Environmental and Health Impacts of Air Pollution: A Review. Front. Public Health, 2020, 8, 1–13.

Information contained in lines 395-422 have been supplemented with the recommended values of the daily intake of chromium and the norms of the content of this element in the blood. It also clarified which oxidation state chromium exhibits toxic properties and on which it is an important part of the diet:

In both the life cycle of the wind power plant and the photovoltaic power plant, substances that are produced are the source of human carcinogenic toxicity. Their higher emission level was recorded for the first of mentioned technical facilities (the total impact at the level of 5.82×103 Pt). For both analyzed systems of renewable energy sources, chromium emissions to water had the greatest impact on the shaping of harmful environmental consequences in this impact category (W – 3.83×103 Pt and PV – 1.23×103 Pt) and atmosphere (W – 1.70×103 Pt and PV – 9.18×101 Pt) (Table 5). Chromium is an element that occurs in two forms - Cr (III) found in food, which is an important component of the diet, and Cr (VI), which is toxic to humans, and its derivatives are used in various industries. Chromium III is an element necessary for the proper functioning of the human body.. It plays a very important role in the processes of insulin action and exerts a significant influence on glucose metabolism and its level in the blood. It takes part in antioxidant processes and plays a role in the functioning of the immune system. It is part of enzymes and is a catalyst for many chemical reactions. Both chromium deficiency and excess can be harmful. The recommended, safe dietary intake of chromium is 50-200 μg per day. If it is exceeded, it has negative consequences for health. The reference values (norms) for the level of chromium in the blood are 1.5-4.7 nmol/l. On the other hand, chromium VI is recognized as being carcinogenic, mutagenic, embryotoxic and teratogenic. It is harmful to the digestive system, respiratory system and skin. It damages the skin and mucous membranes, and inhaling chromium VI compounds can damage the nose, throat, lungs, stomach and intestines. Chronic exposure to chromium VI is associated with its negative effects on the immune, hematological and reproductive systems and the risk of cancer development. Chromium reaches the body through food, inhalation and through the skin. Chromic VI acid salts are used in many industries - in the metallurgical, chemical and construction industries, in the production of pigments, polymers and glass products. For this reason, a significant level of emissions of this element is recorded in the lifecycle of winds and photovoltaic power plants (especially during the production of their materials, materials and components) [62].

References no. 62: Shekhawat, K.; Chatterjee, S.; Joshi, B. Chromium toxicity and its health hazards. Int. J. Adv. Res, 2015, 7, 167–172.

Information on the impact of arsenic on human health:

Lines 426-447: The last of the impact categories, which is characterized by a particularly high level of negative environmental consequences in the life cycles of both considered technical objects, is human non-carcinogenic toxicity. Again, the wind power plant lifecycle has a higher environmental impact in this area (total 3.19×103 Pt) compared to the lifecycle of a photovoltaic power plant (total 1.93×103 Pt). The highest level of emissions in the lifecycle of both renewable energy systems was recorded for arsenic (air: W – 9.46×102 Pt and PV – 4.88×102 Pt, water: W – 8.05×102 Pt and PV – 9.18×102 Pt) and lead (W – 7.45×102 Pt and PV – 2.83×102 Pt) (Table 6). Arsenic is a substance harmful to health. According to the World Health Organization (WHO), it is one of the top 10 chemical compounds of greatest importance to public health. Arsenic negatively affects the enzymatic processes in the cells of the body. It disrupts the work of the nervous, cardiovascular, respiratory and reproductive systems, has an adverse effect on the production of hormones and the body's immunity. Symptoms of arsenic poisoning usually occur after many years of exposure to this element (for example due to its use in industry). Arsenic is not excreted from the body in metabolic processes - it is deposited in it, accumulates and slowly poisons all systems and organs. Human activities contribute to the release of arsenic to the atmosphere, water and soil. The sources of pollution are, among others mining and smelting industry of metal ores and coal combustion. Arsenic is used, for example, in the production of semiconductors and to improve the quality of some metal alloys. This is the cause of significant emissions of this element in the lifecycle of winds and photovoltaic power plants (mainly as part of the production processes of materials and elements of these technical facilities) [63].

References no. 63: Jomova, K.; Jenisova, Z.; Feszterova, M.; Baros, S.; Liska, J.; Hudecova, D.; Rhodes, C.J.; Valko, M. Arsenic: toxicity, oxidative stress and human disease. J. Appl. Toxicol., 2011, 31, 95–107.

Information on the impact of carbon dioxide on the environment and human health:

Lines 469-496: Comparing the lifecycle of wind and photovoltaic power plants, it is clear that the former has more negative effects (total: W – 8.44×104 Pt and PV – 2.90×104 Pt) in terms of the impact on human health. The substances causing the most destructive consequences in the discussed area of influences, in the life cycle of both considered technical renewable energy facilities, include carbon dioxide (W – 2.73×104 Pt and PV – 8.05×103 Pt), particulates < 2.5 mm (W – 2.58×104 Pt and PV – 6.43×103 Pt) and sulfur dioxide (W – 1.33×104 Pt and PV – 7.15×103 Pt) (Table 7). Carbon dioxide occurs in the human body and is produced in it, plays an important role, among others. in maintaining the acid-base balance of the body, oxygen transport and relaxation of smooth muscles in the wall of blood vessels. It is part of the carbon cycle in nature, it is a product of combustion and respiration (hence its significant amount is emitted mainly in the processes of producing materials and elements of wind and photovoltaic power plants, which are characterized by significant energy and material consumption). Excess carbon dioxide in the atmosphere causes, among others, acidifying the water that absorbs it, which is important for many marine ecosystems. Above all, however, an increase in the concentration of this chemical compound enhances the greenhouse effect. This leads not only to an increase in the temperature of the Earth's surface, but also to many other consequences. Increased concentration of carbon dioxide in the inhaled air is one of the important factors that may increase the concentration of CO2 in the blood and cerebrospinal fluid. Its action causes dyspnea, hypercapnia and subsequent cerebral edema. These cases seem to be extremely extreme. Carbon dioxide, however, affects the human body every day and most people felt the negative effects of too high a concentration of this gas in the air. Increased concentration of CO2 disrupts human cognitive processes (from making simple decisions to complex strategic thinking), its concentration achieved after several hours in a closed room has a negative impact on the effectiveness of learning, memory and concentration. Carbon dioxide s a substance without which life on Earth and the functioning of organisms would not be possible, but the problem is not its existence itself, but the increase in its concentration, which occurs at an increasingly faster pace [60–61].

References no. 60: Howarth, R.W.; Jacobson, M.Z. How green is blue hydrogen? Energy Sci Eng. 2021, 9, 1676–1687.

References no. 61: Manisalidis, I.; Stavropoulou, E.; Stavropoulos, A.; Bezirtzoglou, E. Environmental and Health Impacts of Air Pollution: A Review. Front. Public Health, 2020, 8, 1–13.

  1. The section “Summary and conclusions” should be removed. A section “Discussion” should be created where the results will be considered in the light of comparison with the other studies, not only with similar methodology, but in the relation to sustainable development of energy systems at all, …. , discuss the strengths and shortcomings of used method and calculation procedure, etc. If you just get some data and put it in is some software and then show outputs there is no point. Give “deeper” meaning to the results and how much of a negative impact it really is (numbers by themselves mean nothing).

Thank you very much for this attention.

Major changes have been made to the last section of the article.

There are research in the life cycle of wind and photovoltaic power plants in the world literature. However, in Poland, analyzes using the LCA methodology are still not very popular. The study tried to outline the local perspective of the environmental impact of selected renewable energy sources, hence it was decided to carry out a study of two real cases - a 2 MW wind power plant and a 2 MW photovoltaic power plant located in Poland. By maintaining a local perspective, the authors wanted to draw attention to wider issues and contribute to increasing social awareness of the environmental consequences of the life cycle of the considered renewable energy sources. Additionally, it can be noticed that the world literature lacks studies in which analyzes of the life cycle of wind and photovoltaic power plants would be performed with the use of a relatively new method ReCiPe 2016. Most of the research conducted focuses only on the impact of the life cycle of the plant on GWP (Global Warming Potential), ignoring other negative impacts on the quality of the environment and human health and the depletion of raw materials, which also require detailed analyzes, especially in the perspective of sustainable development of energy systems. Therefore, the last section of the article has been renamed from "Summary and conclusions" to "Summary and discussion" and a discussion on the life cycle analyzes of wind and photovoltaic power plants has been added:

Lines 591-647: LCA analyzes in the field of wind energy initially assumed power plants with a capacity of less than one MW as their research subject. Schleisner [67] conducted one of the first studies of this type for a 500 kW turbine, while Ardente et al. [68] performed an analysis for a wind farm consisting of 11 turbines with a capacity of 660 kW each. However, several analyzes were also carried out for wind energy systems with high installed capacity, e.g. Alexandra et al. performed LCA tests for two onshore and two offshore wind power plants [69]. There are also studies devoted to local issues: Martínez et al. [70] studied the impact of the wind power plant life cycle on the environment in Spain, Wagner et al. [71] – in Germany, Schleisner [67] – in Denmark, Ardente et al. [68] – in Italy, Al-Behadili and El-Osta [72] – in Libia, Kabir et al. [73] – in Canada, Alsaleh et al. [74] – in United States, Vargas et al. [75] – in Mexico, and Oebelsa et al. [76] – in Brasil. In the case of this study, local conditions for Poland were taken into account. However, there are very little studies in the world literature in which the analyzes of the life cycle of wind power plants would be performed with the use of a relatively new method ReCiPe 2016. Most of the research conducted focuses only on the impact of the life cycle of the power plant on GWP (Global Warming Potential), ignoring other negative impacts on the quality of the environment and human health and the depletion of raw materials, which also require detailed analyzes, especially in the perspective of sustainable development of energy systems. Kabir et al. [73] examined three models of wind turbines of different power, discovering that the higher the power, the lower the CO2 emissions per kWh of produced energy. Oebels et al. [76] investigated a 1.5 MW power plant and found that the life cycle of its steel tower was mainly responsible for the highest greenhouse gas emissions. Chipindula et al. [77] performed LCA of offshore and onshore wind power plants with different installed capacity, obtaining results confirming that its increase translates into a reduction in carbon dioxide emissions per amount of electricity generated. Going further Alsaleh et al. [74] analyzed a 2 MW turbine, considering different periods of operation of this type of facilities, reaching the conclusion that the production stage is causing the highest GHG emissions to the atmosphere.

In the case of LCA analyzes conducted for photovoltaic systems, a similar trend can be observed as for wind energy systems. The subject of their research are usually various types of materials from which PV modules are produced. The largest number of analyzes was devoted to elements made of silicon, e.g. Alsema [78], Frankl et al. [79], Fthenakis and Kim [80], Dones and Frischknecht [81] also Kato et al. [82] studied the lifecycle of modules made with single-crystalline silicon (sc-Si), and on the other hand the analyzes of Alsema [78], Fthenakis and Alsema [83], Fthenakis and Kim [80], Dones and Frischknecht [81], Ito et al. [84, 85], Kato et al. [82], Nomura et al. [86] also Oliver and Jackson [87] focused on multi-crystalline silicon (mc-Si) while the research conducted by Alsema [78], Ito et al. [85], Kato et al. [82] included amorphous-silicon (a-Si), while Bravi et al. [88] assessed modules made of multi-junction thin-film silicon (µc-Si). In the literature, can also be found works devoted to other materials, e.g., research Fthenakis and Alsema [83], Fthenakis and Kim [80] also Ito et al. [85] were devoted to PV modules with cadmium telluride (CdTe), while Bravi et al. [88] analyzed the modules made from copper-indium-gallium-diselenide (CIGS), Greijer et al. [89] – dye-sensitized solar cells (DSSC). However, several studies were also carried out for high-power photovoltaic systems, e.g., Schaefer and Hagedorn [90] (2.5 MW, cells sc-Si, mc-Si and a-Si), Kato et al. [91] (10, 30 and 100 MW, cells mc-Si and a-Si) or Kato et al. [92] (10, 30 and 100 MW, cells CdTe). As in the case of wind energy power plants, also for photovoltaic power plants you can find studies devoted to local issues: Schaefer and Hagedorn [90] studied the environmental impact of the life cycle of PV systems in Germany, Dones and Frischknecht [81] – in Switzerland, Alsema [78] – in Netherlands, Bravi et al. [88] and Frankl et al. [79] – in Italy, Fthenakis and Kim [80] – in United States, Kato et al. [82, 91, 92] and Nomura et al. [86] – in Japan, Ito at el. [84, 85] – in China. In the world literature on the assessment of the life cycle of photovoltaic power plants, there are very little studies in which the analyzes would be performed using the method ReCiPe 2016. Most of the research conducted is focused on assessing the amount of CO2 and other greenhouse gas emissions [7892]. Other impacts lowering the quality of ecosystems, posing a threat to human health and exacerbating the depletion of raw materials resources, are usually not taken into account.

  1. The meaning of the unit “Pt” was given poorly. Use term “ecopoint” instead “environmental point”.

Thank you for attention.

The definition of a unit has been expanded “Pt”:

Lines 262-268: Carrying out the weighting process, allowed to obtain the results in ecopoints (Pt). Ecopoint is a unit of measure for the environmental impact of an individual, process, material, element or product. The results presented in ecopoints reflect the influence of the average European on the environment. One thousand ecopoints are equal to the environ-mental impact of one European in one year. The more ecopoints a given unit, process, material, element or product has, the greater its negative environmental impact.

Throughout the article, the term "environmental point" has been replaced with "ecopoint", e.g.:

Lines 262-263: Carrying out the weighting process, allowed to obtain the results in ecopoints (Pt).

Lines 263-268: Ecopoint is a unit of measure for the environmental impact of an individual, process, material, element or product. The results presented in ecopoints reflect the influence of the average European on the environment. One thousand ecopoints are equal to the environ-mental impact of one European in one year. The more ecopoints a given unit, process, material, element or product has, the greater its negative environmental impact.

Lines 284-285: They present the grouping and weighting results in the unit ecopoints (Pt) (discussed in section 2.4).

Round 2

Reviewer 2 Report

The simulation results are provided to show the efficiency of the proposed algorithm. Some comments are listed below.

1. the motivation of the paper is not clear. The author should rewrite this part in the introduction section.

3. figures needs more description.

4. The table needs a reference.

5. Some equations need references.

Author Response

RESPONSE TO REVIEWER 2 COMMENTS

Dear Reviewer,

Thank you for taking the time to read our article and for the review. We appreciate all comments and suggestions as they will help us improve our research workflow in the future.

 Detailed answers to all comments are provided below. Amendments in the text of the article are marked in red.

  1. The motivation of the paper is not clear. The author should rewrite this part in the introduction section.

Thank you very much for this valuable remark.

The summary briefly presents the most important reasons for undertaking research in the field of assessing the life cycle of wind and photovoltaic power plant:

Lines 16-23: The conversion of kinetic energy of wind and solar radiation into electricity during the operation of wind and photovoltaic power plants causes practically no emissions of chemical compounds harmful to the environment. However, the production of their materials and components, and then their post-use management after the end of their operation, are very energy and material consuming. For this reason, this article aims to assess the lifecycle of a wind and photovoltaic power plant in the context of the sustainable development of energy systems. The object of the research were two actual technical facilities - 2 MW wind power plant and 2 MW photovoltaic power plant, located in Poland.

In the introduction, an attempt was made to discuss the above-mentioned causes, referring to the problem of increasing demand for energy from more sustainable sources and the related climate changes, caused by a high share of conventional energy sources in the energy mixes of most countries of the world:

Lines 40-46: Year by year, the world needs much more energy, including powering houses, industrial machines or transport, due to the constantly growing population and ever higher living standards. However, in order to counteract climate change, energy must increasingly come from more sustainable sources, with lower emissions of harmful substances into the environment. Thanks to the knowledge, technology and innovation, man is able to generate more "cleaner" energy. Life and livelihoods, economies and communities depend on a convenient, reliable and affordable energy supply [1–4].

Lines 47-52: Most of the energy used today comes from crude oil and coal, which are non-renewable energy sources. By 2050, the world's population is expected to grow to 9 billion (almost 2 billion more than today). Many people in developing economies will join the global middle class. They will buy various types of machinery and equipment that will consume significant amounts of energy. Global energy demand could double by year 2050 compared to levels from year 2000 [5–7].

The need to introduce changes to the energy system was also mentioned in order to reduce the emission of harmful compounds to the environment and make it more sustainable:

Lines 53-61: Therefore, it is extremely important to counteract climate change caused by emissions of harmful substances and other destructive effects on the condition of the natural environment. In order to meet these challenges, radical changes in the global energy system and a number of new energy sources are needed. Fossil energy sources will continue to play an important role in the decades to come, but the use of alternative and innovative technologies will increasingly contribute to meeting the world's growing energy needs and allow for more efficient, sustainable ways of using energy. These activities should be based primarily on cooperation, respect for the environment and social responsibility [8–11].

It was also noted that each energy source, also renewable, has a certain impact on the environment. Renewable energy sources are considered the most "friendly" to the environment, but their life cycle is also associated with a specific demand for energy and matter, as well as some negative environmental consequences. Therefore, it was considered necessary to look more closely at the environmental impacts occurring in their life cycle in order to limit the harmful effects and to make efforts to make them more sustainable:

Lines 62-71: Each source of energy has a certain effect on the environment. Renewable energy sources are considered to be the most environmentally-friendly sources of energy, those causing the least negative impact that is. Their exploitation is primarily aimed at slowing down climate change. They are a solution for global corporations, local entrepreneurs and individual consumers. More and more countries are investing in alternative energy sources and supporting their development, for example through subsidy programs or low-interest loans. One of the most popular renewable energy sources in the world are solar and wind energy installations. However, the life cycle of machines and devices, including those of renewable energy, is related to a specific demand for matter and energy [12–15].

In addition, it was decided to more precisely justify the choice of the LCA method as a research tool:

Lines 72-88: Sustainable development is about finding solutions that guarantee further economic growth, which allow for the active inclusion of all social groups in development processes, while giving them the opportunity to benefit from this growth. Initially, it was understood as the need to reduce the negative impact of economies on the natural environment. Over the years, the concept has acquired a more complete understanding, aligning the essence of the three development factors: respect for the environment, social progress and economic growth. One of the most popular methods used in analyzes in the area of sustainable development is LCA - Life Cycle Assessment. This method allows to assess the potential impact on the environment of both products and processes in the perspective of their entire life cycle ('from cradle to grave') – starting from the extraction of raw materials, through production, exploitation, ending with post-consumer management. Because of this, no stage of the product life cycle is skipped. As a result of the identification and quantitative assessment of the existing environmental loads, it is possible to analyze the potential impact of these loads on the environment, and consequently – developing recommendations to reduce their negative impacts over the entire life cycle. LCA is a flexible method that allows for individual adjustment of the purpose and scope of the research of the analyzed object of analysis [16–19].

There are research in the life cycle of wind and photovoltaic power plants in the world literature. However, in Poland, analyzes using the LCA methodology are still not very popular. The study tried to outline the local perspective of the environmental impact of selected renewable energy sources, hence it was decided to carry out a study of two real cases - a 2 MW wind power plant and a 2 MW photovoltaic power plant located in Poland. By maintaining a local perspective, the authors wanted to draw attention to wider issues and contribute to increasing social awareness of the environmental consequences of the life cycle of the considered renewable energy sources. Additionally, it can be noticed that the world literature lacks studies in which analyzes of the life cycle of wind and photovoltaic power plants would be performed with the use of a relatively new method ReCiPe 2016. Most of the research conducted focuses only on the impact of the life cycle of the plant on GWP (Global Warming Potential), ignoring other negative impacts on the quality of the environment and human health and the depletion of raw materials, which also require detailed analyzes, especially in the perspective of sustainable development of energy systems.

A broader discussion of the authors' motivations to undertake the analyzes carried out as part of this article has been added:

Lines 89-102: There are not many studies in the world literature in which analyzes of the life cycle of wind and photovoltaic power plants would be performed using a relatively new method ReCiPe 2016. Most of the research conducted focuses only on the impact of the life cycle on GWP (Global Warming Potential), ignoring other negative impacts of the systems under consideration, which reduce the quality of the environment, pose a threat to human health and increase the depletion of raw materials, which also require detailed analyzes, especially in view of the sustainable development of energy systems (see section 5 for details). In Poland, unfortunately, analyzes using the LCA methodology are still not very popular. The study tries to outline the local perspective of the environmental impact of selected renewable energy sources, hence it was decided to study two real cases.

Therefore, the main objective of this study is to assess the life cycle of a wind and photovoltaic power plant in the context of sustainable development of energy systems. It will be based on a study of two real cases - a 2 MW wind power plant and a 2 MW photo-voltaic power plant located in Poland.

We hope that we have managed to explain the motivation for undertaking research in the field of life cycle assessment of a selected wind and photovoltaic power plant.

  1. Figures needs more description.

Thank you for your attention.

Corrections in descriptions of figures has been added:

Lines 105-116: The life cycle assessment was carried out for the onshore 3-blade 2 MW horizontal wind power plant located in central Poland and the photovoltaic power plant with silicon monocrystalline photovoltaic panels (without PV tracking system), with a capacity of 2 MW, located in the northern part of Poland. Life cycle assessment of materials and elements of renewable energy systems is possible thanks to the use of various models, including Environmental LCA. This method has been chosen as the model of assessing the potential impact wind and photovoltaic power plants on human health, ecosystems quality and resource depletion. In accordance with ISO 14040 (Environmental management, Life cycle assessment, Principles and framework) and ISO 14044 (Environmental management, Life cycle assessment, Requirements and guidelines) standards, the LCA analysis performed in this work included four stages: determination of goal and scope, life cycle inventory (LCI), life cycle impact assessment (LCIA) and interpretation (Figure 1) [2023].

Lines 172-191: The total mass of materials and elements of the tested wind power plant is about two thousand tons. Foundations have the largest share in the mass of the object - approx. 79% (of which approx. 96% is concrete, and the remaining 4% - steel). The other most important elements of the analyzed power plant include the tower with approx. 15% share in the weight of the entire facility (mostly made of steel), the rotor with approx. 2% (including about half of this mass are blades made of polymers reinforced with fiberglass, and the other half - hub made mainly of nodular cast iron) and nacelle with approx. 4% share (elements of which are mainly made of cast iron - approx. 49 % by weight of nacelle, steel - approx. 38%, aluminum - approx. 4%, polymer materials - approx. 3%, copper - approx. 2%) (Figure 2 and 3) (data obtained from manufacturers) (data obtained from the investor and producers).

The total mass of materials and elements of the tested photovoltaic power plant is around three hundred tons. Single-crystalline silicon photovoltaic modules have the largest share in the mass of the object - approx. 62% (including approx. 47% of their mass is solar glass, and approx. 45% - aluminum). The other most important elements of the analyzed power plant include supporting structure with approx. 21% share in the weight of the entire facility (mostly made of steel), inverter station with approx. 15% share (elements of which are mainly made of steel - approx. 42% and aluminum - approx. 38%) and electrical installation with approx. 2% share (mostly made of copper) (Figure 2 and 3) (data obtained from the investor and producers).

Lines 199-211: When determining the impact of the life cycle of a given technical facility on the environment, the third phase of the analysis - life cycle impact assessment - is of key importance. Any methodological differences in the LCA approaches mainly relate to the LCIA phase, which is complex with mandatory and optional elements. Mandatory elements include selection of impact categories, category indicators, characterization models, classification and characterization, and the optional elements include normalization, grouping and weighting (Figure 4). The mandatory elements sequence is strictly defined and must be preserved for parsing. The question of choice is whether and which optional elements will be used. As part of the research, all the listed optional elements were used (normalization, grouping and weighting). The analyzes under this study were carried out using the SimaPro 9.3 software (PRé Sustainability, LE Amersfoort, Netherlands) with Ecoinvent 3.8 database. The life cycle assessment of wind and photovoltaic power plants was carried out using the method ReCiPe 2016 [20,37–40].

Lines 215-224: ReCiPe is one of the methods used in the stage life cycle impact assessment (LCIA). It was first developed in 2008 through cooperation between the Dutch National Institute for Public Health and the Environment (RIVM), Radboud University Nijmegen, Leiden University and PRé Sustainability. The main purpose of the ReCiPe method is to convert life cycle inventory results into indicator scores. Indicator scores express the potential magnitude of the impact on the environment impact category. Under the ReCiPe method, indicators are determined on two levels – 22 midpoint indicators (midpoint impact category) and 3 endpoint indicators (endpoint area of influence). Midpoint indicators focus on single environmental problem, in turn endpoint indicators – show the environmental impact on three higher aggregation levels (Figure 5) [41–42].

Lines 452-459: In the ReCiPe 2016 method, the results of the 22 impact categories indicators are summed up into three areas of influence - human health, ecosystems and resources. The highest level of harmful impact in the case of wind power plants (jointly 8.44×104 Pt), and photovoltaic power plant (jointly 2.90×104 Pt) noted for areas of influence related to the impact on human health, in turn, the lowest - it was characterized by the impact in the area of processes related to the depletion of resources. In the case of impact on environmental quality, the life cycle of a wind power plant had more negative consequences compared to a photovoltaic power plant (Figure 6).

Lines 463-468: In the lifecycle of both analyzed technical objects, the impact on human health accounts for approx. 94% of all negative consequences in relation to the environment (for ecosystems it is approx. 5%, and for resources - approx. 1%). The triangle view also shows a higher level of harmful effects on the milieu generated throughout the life cycle of a wind power plant (blue color in the chart), compared to the life cycle of a photovoltaic power plant (red color in the chart) (Figure 7).

Lines 507-517: Among the most important types of emission of harmful substances to the environment, three can be distinguished - to the atmospheric, water and soil environment. In the case of both analyzed life cycles, emissions to the atmosphere cause the most negative environmental consequences ( total: W – 8.36×104 Pt and PV – 2.83×104 Pt). For the lifetime of the wind power plant, emissions to the atmosphere account for over 94% of all emissions (water: approx. 6%, soil: < 1%), on the other hand, for the life cycle of the photovoltaic power plant - approx. 93% (water: approx. 7%, soil: < 1%). Emissions to the atmospheric environment in both life cycles of the analyzed technical facilities cause the most negative consequences in the area of deteriorating human health (W – 7.93×104 Pt and PV – 2.68×104 Pt). For this reason, it was decided to analyze the human health area of influence in more detail later in section 3.3 (Figure 8).

  1. The table needs a reference.

Sincerely thank you for this remark.

All tables presented in the article contain the results of own research. For this reason, the references (relating, for example, to the effects of a substance on human health or the quality of the environment) are only included in the description of the tables.

  1. Some equations need references.

Thank you very much for attention.

There is no equation in this article.

Reviewer 4 Report

No additional comments.

Author Response

RESPONSE TO REVIEWER 4 COMMENTS

Dear Reviewer,

Thank you for taking the time to read our article and for the review. We appreciate all  previous comments and suggestions as they will help us improve our research workflow in the future.

Round 3

Reviewer 2 Report

1. There is no need to subsections in the introduction. Please merge all subsections in the introduction as one section.  

2. Please add a reference(s) to the equations as can as possible.

3. Please add a table of nomenclatures or define each symbol directly under the equations.

4. If possible, add a new flowchart to show the steps of the simulation.

5. Could you explain how can verify the results?

Author Response

RESPONSE TO REVIEWER 2 COMMENTS

Dear Reviewer,

Thank you for taking the time to read our article and for the review. We appreciate all comments and suggestions as they will help us improve our research workflow in the future.

 Detailed answers to all comments are provided below.

  1. There is no need to subsections in the introduction. Please merge all subsections in the introduction as one section.

Thank you very much for this remark.

There is no subsection in the introduction:

Lines 39-102:

  1. Introduction

Year by year, the world needs much more energy, including powering houses, industrial machines or transport, due to the constantly growing population and ever higher living standards. However, in order to counteract climate change, energy must increasingly come from more sustainable sources, with lower emissions of harmful substances into the environment. Thanks to the knowledge, technology and innovation, man is able to generate more "cleaner" energy. Life and livelihoods, economies and communities depend on a convenient, reliable and affordable energy supply [14].

Most of the energy used today comes from crude oil and coal, which are non-renewable energy sources. By 2050, the world's population is expected to grow to 9 billion (almost 2 billion more than today). Many people in developing economies will join the global middle class. They will buy various types of machinery and equipment that will consume significant amounts of energy. Global energy demand could double by year 2050 compared to levels from year 2000 [57].

Therefore, it is extremely important to counteract climate change caused by emissions of harmful substances and other destructive effects on the condition of the natural environment. In order to meet these challenges, radical changes in the global energy system and a number of new energy sources are needed. Fossil energy sources will continue to play an important role in the decades to come, but the use of alternative and innovative technologies will increasingly contribute to meeting the world's growing energy needs and allow for more efficient, sustainable ways of using energy. These activities should be based primarily on cooperation, respect for the environment and social responsibility [811].

Each source of energy has a certain effect on the environment. Renewable energy sources are considered to be the most environmentally-friendly sources of energy, those causing the least negative impact that is. Their exploitation is primarily aimed at slowing down climate change. They are a solution for global corporations, local entrepreneurs and individual consumers. More and more countries are investing in alternative energy sources and supporting their development, for example through subsidy programs or low-interest loans. One of the most popular renewable energy sources in the world are solar and wind energy installations. However, the life cycle of machines and devices, including those of renewable energy, is related to a specific demand for matter and energy [1215].

Sustainable development is about finding solutions that guarantee further economic growth, which allow for the active inclusion of all social groups in development processes, while giving them the opportunity to benefit from this growth. Initially, it was understood as the need to reduce the negative impact of economies on the natural environment. Over the years, the concept has acquired a more complete understanding, aligning the essence of the three development factors: respect for the environment, social progress and economic growth. One of the most popular methods used in analyzes in the area of sustainable development is LCA - Life Cycle Assessment. This method allows to assess the potential impact on the environment of both products and processes in the perspective of their entire life cycle ('from cradle to grave') – starting from the extraction of raw materials, through production, exploitation, ending with post-consumer management. Because of this, no stage of the product life cycle is skipped. As a result of the identification and quantitative assessment of the existing environmental loads, it is possible to analyze the potential impact of these loads on the environment, and consequently – developing recommendations to reduce their negative impacts over the entire life cycle. LCA is a flexible method that allows for individual adjustment of the purpose and scope of the research of the analyzed object of analysis [1619].

There are not many studies in the world literature in which analyzes of the life cycle of wind and solar power plants would be performed using a relatively new method ReCiPe 2016. Most of the research conducted focuses only on the impact of the life cycle on GWP (Global Warming Potential), ignoring other negative impacts of the systems under consideration, which reduce the quality of the environment, pose a threat to human health and increase the depletion of raw materials, which also require detailed analyzes, especially in view of the sustainable development of energy systems (see section 5 for details). In Poland, unfortunately, analyzes using the LCA methodology are still not very popular. The study tries to outline the local perspective of the environmental impact of selected renewable energy sources, hence it was decided to study two real cases.

Therefore, the main objective of this study is to assess the life cycle of a wind and photovoltaic power plant in the context of sustainable development of energy systems. It will be based on a study of two real cases - a 2 MW wind power plant and a 2 MW photovoltaic power plant located in Poland.

  1. Please add a reference(s) to the equations as can as possible.

Thank you for your attention.

There is no equation in this article.

  1. Please add a table of nomenclatures or define each symbol directly under the equations.

Sincerely thank you for this remark.

There is no equation in this article.

  1. If possible, add a new flowchart to show the steps of the simulation.

Thank you very much for attention.

There is already flowchart showing the steps of the simulation:

Lines 104-133:

2.1. Object and Plan of Analysis

The life cycle assessment was carried out for the onshore 3-blade 2 MW horizontal wind power plant located in central Poland and the photovoltaic power plant with silicon monocrystalline photovoltaic panels (without PV tracking system), with a capacity of 2 MW, located in the northern part of Poland. Life cycle assessment of materials and elements of renewable energy systems is possible thanks to the use of various models, including Environmental LCA. This method has been chosen as the model of assessing the potential impact wind and photovoltaic power plants on human health, ecosystems quality and resource depletion. In accordance with ISO 14040 (Environmental management, Life cycle assessment, Principles and framework) and ISO 14044 (Environmental management, Life cycle assessment, Requirements and guidelines) standards, the LCA analysis performed in this work included four stages: determination of goal and scope, life cycle inventory (LCI), life cycle impact assessment (LCIA) and interpretation (Figure 1) [2023].

(Please see the attachment)

Figure 1. The stages of the LCA analysis (in accordance with ISO 14040 and 14044 standards).

The research was started with the description of goal and scope (details are provided in section 2.2). Based on an earlier analysis of the state of the art and technology, it was found that the literature lacks a detailed life cycle assessment of wind and photovoltaic power plants in the context of the sustainable development of energy systems. It was also extremely important when formulating the goal and scope to collect as many, and possibly the best qualitatively, data on the objects of analysis. It was possible thanks to cooperation with companies producing materials and elements of wind and photovoltaic power plants, which have a leading position on the European and domestic market. A more detailed description of the second part of the research (LCI) is provided in section 2.3. In the next step, a detailed analysis of the life cycle of the considered technical objects was made. The necessary simulation analyzes were carried out using the SimaPro 9.3 software, using the ReCiPe 2016 calculation procedure. The course of this stage was presented in section 2.4, and the obtained results and their interpretation - in section 3. The last part of the study (described in section 2.5) included the interpretation of the obtained results and is presented in sections 3 and 4 [20,2427].

Additionally, section 2.4 describes in detail the theoretical background and the plan for the steps of the simulation:

Lines 199-270:

When determining the impact of the life cycle of a given technical facility on the environment, the third phase of the analysis - life cycle impact assessment - is of key importance. Any methodological differences in the LCA approaches mainly relate to the LCIA phase, which is complex with mandatory and optional elements. Mandatory elements include selection of impact categories, category indicators, characterization models, classification and characterization, and the optional elements include normalization, grouping and weighting (Figure 4). The mandatory elements sequence is strictly defined and must be preserved for parsing. The question of choice is whether and which optional elements will be used. As part of the research, all the listed optional elements were used (normalization, grouping and weighting). The analyzes under this study were carried out using the SimaPro 9.3 software (PRé Sustainability, LE Amersfoort, Netherlands) with Ecoinvent 3.8 database. The life cycle assessment of wind and photovoltaic power plants was carried out using the method ReCiPe 2016 [20,3740].

(Please see the attachment)

Figure 4. The elements of a life cycle impact assessment (in accordance with ISO 14040 and 14044 standards).

ReCiPe is one of the methods used in the stage life cycle impact assessment (LCIA). It was first developed in 2008 through cooperation between the Dutch National Institute for Public Health and the Environment (RIVM), Radboud University Nijmegen, Leiden University and PRé Sustainability. The main purpose of the ReCiPe method is to convert life cycle inventory results into indicator scores. Indicator scores express the potential magnitude of the impact on the environment impact category. Under the ReCiPe method, indicators are determined on two levels – 22 midpoint indicators (midpoint impact category) and 3 endpoint indicators (endpoint area of influence). Midpoint indicators focus on single environmental problem, in turn endpoint indicators – show the environmental impact on three higher aggregation levels (Figure 5) [4142].

ReCiPe 2016 includes the broadest set of midpoint impact categories compared to other methods. Unlike other approaches (for example Eco-indicator 99 or Impact 2002+) it doesn’t include potential impacts from future extractions in the impact assessment but assumes such impacts have been included in the inventory analysis. The ReCiPe 2016 method is an improvement of ReCiPe 2008 and previously used methods such as CML 2000 and Eco-indicator 99. In contrast with the previous version, ReCiPe2016 also provides global characterization factors instead of only European [4344].

(Please see the attachment)

Figure 5. Overview of the impact categories that are covered in the ReCiPe2016 method and their relation to the areas of influence.

Assigning LCI results to individual impact categories is referred to as classification. The use of appropriate, specialized software makes it possible to automate this stage. The SimaPro program was used for classification, which automatically assigns LCI results to individual impact categories, based on a list of substances belonging to given calculation methods and databases included in the program [20,4546].

Characterization and conversion of LCI results into the results of impact categories indicators are extremely complex processes. From a technical point of view, they come down to converting the LCI results through appropriate characterization parameters and showing in the form of relative shares in each of the impact categories. The main calculation procedure used in this analysis was the ReCiPe 2016 method [20,4748].

Normalization is understood as computing the magnitude of the results of a category indicator against reference information. It is used to determine the relative importance of the indicator results compared to a given region (like Poland or Europe) or a person (for example an average inhabitant of Poland, Europe) in a specific period of time. Normalization can also be used to prepare LCIA results for subsequent procedures, for example weighting. As part of the research, normalization was performed with the use of the SimaPro software. It was a necessary stage to carry out the next steps - grouping and weighting [20,4950].

There are different evaluation methods and preferences for impact categories. Depending on the goal and scope of the analysis, some may be more important than others. Additionally, they can be grouped, for instance according to the emission level or scale (local, global). In the ReCiPe 2016 method, grouping takes place when the results of the impact categories indicators are summed up into three areas of influence and during the final aggregation to the total impact indicator [5152].

The weighting process consists in determining and assigning a degree of importance (weighting factor) to individual impact categories and multiplication by the normalized index results. Weighting should be performed on complete, internationally recognized sets of weighting factors that have been developed for all impact categories. Carrying out the weighting process, allowed to obtain the results in ecopoints (Pt). Ecopoint is a unit of measure for the environmental impact of an individual, process, material, element or product. The results presented in ecopoints reflect the influence of the average European on the environment. One thousand ecopoints are equal to the environmental impact of one European in one year. The more ecopoints a given unit, process, material, element or product has, the greater its negative environmental impact. During the lifecycle analyzes of wind and photovoltaic power plants, grouping and weighting were performed using the SimaPro program, and their results are presented in section 3 [20,5354].

  1. Could you explain how can verify the results?

Thank you very much for this valuable remark.

The interpretation phase aims at discussing the results of the LCI and LCIA, and drawing conclusions, and suggestions. According to ISO, it involves evaluation checks for the trustworthiness of the LCA results by assessing the completeness of data taken into account, and consistency check concerning the goal and scope.

The issue of data quality (in general) is discussed in section 2.3:

Lines 165-168:

Information on key processes was obtained directly from manufacturers of materials and components. Data on less significant processes and materials from the point of view of environmental impact were obtained from databases included in the SimaPro 9.3 software (Ecoinvent 3.8 database).

In accordance with ISO standards, the obtained data was accurate and precise, was complete and obtained from sources deemed reliable by the authors.

Information on the interpretation of the results is provided in section 2.5:

Lines 272-282:

On the one hand, the interpretation is the final part of the LCA analysis (fourth phase), and on the other hand, the interpretation process is still present for each of the three earlier stages of the procedure (determination of goal and scope, LCI and LCIA). The key purpose of the interpretation is the analysis of the results and their verification from the point of view of the previously established goal and scope [20,5556].

Completeness of the analysis was checked with a positive result. The data needed for the interpretation was complete. A compliance check was also carried out during the conducted research. The adopted assumptions, methods used, the depth of the analysis, its detail and precision of data for both materials and wind and photovoltaic power plants elements were consistent with the goal and scope of the research. The obtained results and their interpretation are presented in sections 3 and 4 [5758].

The last section "Summary and discussion" includes a literature review and a discussion of the obtained results, including 26 literature items (items 67-92 in the list of references at the end of the article), in the field of life cycle analysis of wind energy and photovoltaic systems. By comparing the obtained results with the results published in other scientific articles, it was possible to verify their correctness:

Lines 591-647:

LCA analyzes in the field of wind energy initially assumed power plants with a capacity of less than one MW as their research subject. Schleisner [67] conducted one of the first studies of this type for a 500 kW turbine, while Ardente et al. [68] performed an analysis for a wind farm consisting of 11 turbines with a capacity of 660 kW each. However, several analyzes were also carried out for wind energy systems with high installed capacity, e.g. Alexandra et al. performed LCA tests for two onshore and two offshore wind power plants [69]. There are also studies devoted to local issues: Martínez et al. [70] studied the impact of the wind power plant life cycle on the environment in Spain, Wagner et al. [71] – in Germany, Schleisner [67] – in Denmark, Ardente et al. [68] – in Italy, Al-Behadili and El-Osta [72] – in Libia, Kabir et al. [73] – in Canada, Alsaleh et al. [74] – in United States, Vargas et al. [75] – in Mexico, and Oebelsa et al. [76] – in Brasil. In the case of this study, local conditions for Poland were taken into account. However, there are very little studies in the world literature in which the analyzes of the life cycle of wind power plants would be performed with the use of a relatively new method ReCiPe 2016. Most of the research conducted focuses only on the impact of the life cycle of the power plant on GWP (Global Warming Potential), ignoring other negative impacts on the quality of the environment and human health and the depletion of raw materials, which also require detailed analyzes, especially in the perspective of sustainable development of energy systems. Kabir et al. [73] examined three models of wind turbines of different power, discovering that the higher the power, the lower the CO2 emissions per kWh of produced energy. Oebels et al. [76] investigated a 1.5 MW power plant and found that the life cycle of its steel tower was mainly responsible for the highest greenhouse gas emissions. Chipindula et al. [77] performed LCA of offshore and onshore wind power plants with different installed capacity, obtaining results confirming that its increase translates into a reduction in carbon dioxide emissions per amount of electricity generated. Going further Alsaleh et al. [74] analyzed a 2 MW turbine, considering different periods of operation of this type of facilities, reaching the conclusion that the production stage is causing the highest GHG emissions to the atmosphere.

In the case of LCA analyzes conducted for photovoltaic systems, a similar trend can be observed as for wind energy systems. The subject of their research are usually various types of materials from which PV modules are produced. The largest number of analyzes was devoted to elements made of silicon, e.g. Alsema [78], Frankl et al. [79], Fthenakis and Kim [80], Dones and Frischknecht [81] also Kato et al. [82] studied the lifecycle of modules made with single-crystalline silicon (sc-Si), and on the other hand the analyzes of Alsema [78], Fthenakis and Alsema [83], Fthenakis and Kim [80], Dones and Frischknecht [81], Ito et al. [84, 85], Kato et al. [82], Nomura et al. [86] also Oliver and Jackson [87] focused on multi-crystalline silicon (mc-Si) while the research conducted by Alsema [78], Ito et al. [85], Kato et al. [82] included amorphous-silicon (a-Si), while Bravi et al. [88] assessed modules made of multi-junction thin-film silicon (µc-Si). In the literature, can also be found works devoted to other materials, e.g., research Fthenakis and Alsema [83], Fthenakis and Kim [80] also Ito et al. [85] were devoted to PV modules with cadmium telluride (CdTe), while Bravi et al. [88] analyzed the modules made from copper-indium-gallium-diselenide (CIGS), Greijer et al. [89] – dye-sensitized solar cells (DSSC). However, several studies were also carried out for high-power photovoltaic systems, e.g., Schaefer and Hagedorn [90] (2.5 MW, cells sc-Si, mc-Si and a-Si), Kato et al. [91] (10, 30 and 100 MW, cells mc-Si and a-Si) or Kato et al. [92] (10, 30 and 100 MW, cells CdTe). As in the case of wind energy power plants, also for photovoltaic power plants you can find studies devoted to local issues: Schaefer and Hagedorn [90] studied the environmental impact of the life cycle of PV systems in Germany, Dones and Frischknecht [81] – in Switzerland, Alsema [78] – in Netherlands, Bravi et al. [88] and Frankl et al. [79] – in Italy, Fthenakis and Kim [80] – in United States, Kato et al. [82, 91, 92] and Nomura et al. [86] – in Japan, Ito at el. [84, 85] – in China. In the world literature on the assessment of the life cycle of photovoltaic power plants, there are very little studies in which the analyzes would be performed using the method ReCiPe 2016. Most of the research conducted is focused on assessing the amount of CO2 and other greenhouse gas emissions [7892]. Other impacts lowering the quality of ecosystems, posing a threat to human health and exacerbating the depletion of raw materials resources, are usually not taken into account.

Life-cycle interpretation, the last phase of the LCA process, is a systematic technique to identify, quantify, check, and evaluate information from the results of the LCI and the LCIA, and communicate them effectively. In this article, checks are performed regarding completeness of the significant data elements, and consistency (with regard to the goal definition and scope of the study). Such checks are required to reach consistent and reliable conclusions.

Round 4

Reviewer 2 Report

---------